# Topological field theory approach to intermediate statistics

**Ward L. Vleeshouwers[1,2]\* and Vladimir Gritsev[1,3]**

**1** Institute for Theoretical Physics, Universiteit van Amsterdam
**2** Institute for Theoretical Physics, Universiteit Utrecht
**3** Russian Quantum Center, Skolkovo, Moscow, Russia

\* w.l.vleeshouwers@uva.nl

## Abstract

Random matrix models provide a phenomenological description of a vast variety of physical phenomena. Prominent examples include the eigenvalue statistics of quantum (chaotic) systems, which are characterized by the spectral form factor (SFF). Here, we calculate the SFF of unitary matrix ensembles of infinite order with the weight function satisfying the assumptions of Szegö's limit theorem. We then consider a parameter-dependent critical ensemble which has intermediate statistics characteristic of ergodic-to-nonergodic transitions such as the Anderson localization transition. This same ensemble is the matrix model of $U(N)$ Chern-Simons theory on $S^3$, and the SFF of this ensemble is proportional to the HOMFLY invariant of $(2n, 2)$-torus links with one component in the fundamental and one in the antifundamental representation. This is one example of a large class of ensembles with intermediate statistics arising from topological field and string theories. Indeed, the absence of a local order parameter suggests that it is natural to characterize ergodic-to-nonergodic transitions using topological tools, such as we have done here.


# 1 Introduction

## 1.1 Random Matrix Theory in disordered and complex systems: brief overview

The idea of Wigner [1] to describe complex physical systems by treating its Hamiltonian matrix as random has found since then a wide variety of applications. One of the main interests and challenges of modern theoretical physics to which random matrix theory has been very successfully applied is the description of interacting many-particle systems subject to a certain degree of randomness. Physically, this randomness is often caused by a true physical disorder, originating for instance from irregularities in a crystal lattice or by the presence of impurities. One can also have auxiliary phenomenological randomness representing the fact that the interactions in the system are too complicated to be described in microscopic detail, which is the case, for instance, for heavy nuclei. Further, quantum noise induced when a system is in contact with an external bath is a source of a temporal randomness. Random matrix theory (RMT) allows one to deal with such problems on a phenomenological level. This theory cannot answer questions about the microscopic details of a system, but it focuses instead on *universal* relations and scaling properties of relevant quantities. Indeed, one of the main results of RMT is the existence of universality classes (see [2] a for survey), in which the symmetry of the system determines the class and, consequently, the statistical properties of the energy spectrum.

RMT models disordered and/or complicated Hamiltonians as matrices with random elements distributed according to a certain probability. Certain general physical symmetries (like time-reversal symmetry) provide restrictions on how the matrix elements are correlated. This leads to a different classes of random matrices [3], see the classic book by Mehta [4] and a contemporary overview of RMT by Forrester [5]. Here, we will consider ensembles of Hermitian or unitary matrices, in particular, their eigenvalue statistics. A prominent RME is the GUE, which is an ensemble of Hermitian random matrices **H** with Gaussian weight function. This

entails that its eigenvalues are distributed according to a $U(N)$-invariant Gaussian probability distribution $P(H) \sim \exp[-\alpha \text{Tr} V(\mathbf{H})]$, where $V(\mathbf{H}) = \mathbf{H}^2$ and $\alpha$ is a real positive parameter. Other classes correspond to ensembles of real symmetric matrices, with the probability measure being invariant under orthogonal transformations, or self-dual Hermitian matrices with probability distribution invariant under symplectic transformations, known as GOE and GSE, respectively [4]. Another notable generalization is the notion of circular Random matrix Ensemble (RME), where the eigenvalues are distributed across the complex unit circle instead of the real line. The circular analogues of GOE, GUE, and GSE are known as COE, CUE, and CSE, respectively. We will only be considering unitary ensembes here. Further, although many properties are common to the Gaussian and circular ensembles, certain objects are easier to calculate in the circular case, which is why these ensembles are the focus of this paper.

For *typical* systems, which obey a so-called Eigenstate Thermalization Hypothesis (see [6], [7] for a recent review), almost every energy level contains "seeds" of thermal behavior (even for isolated systems) leading to the chaotic nature of the RMT statistics. Therefore, quantum states belonging to this type are called *ergodic*. In disordered systems, the delocalized or chaotic phase is described by Wigner-Dyson statistics, in which case the level spacing distribution is given by $p(s) \sim s^{\beta} e^{a_{\beta} s^2}$, where $s$ is the difference between consecutive energy levels, $\beta = 1, 2, 4$ for the unitary, orthogonal and symplectic cases respectively and $a_{\beta}$ is a constant. As the strength of randomness increases, there can occur a transition to the situation where states of a system are localized in *some* basis. This could be a basis of states relevant for the description of localization in real space (Anderson localization) or in the Hilbert space (many-body localization). Deep inside a localized phase, the behavior of the system is nonergodic and the RMT level's statistics follows a Poisson distribution, $p(s) \sim e^{-s}$. This type of statistics is usually found in quantum *integrable* systems, where a sufficient number of conserved charges significantly constrains the dynamics.

## 1.2   Intermediate statistics and corresponding RMT approaches

Quantum systems whose classical counterparts are somewhere in between ordered and chaotic have spectral statistics that exhibit a mixture of Wigner-Dyson and Poissonian features, which we will refer to as *intermediate statistics*. An important example of such a system is given by disordered conductors, where increasing the disorder strength beyond a certain point causes electrons to undergo Anderson localization. In a sufficiently small energy window around the mobility edge, electrons exhibit the aforementioned intermediate statistics [8]. Further, at the point of transition between extended and localized regimes the wave functions are *multifractal*, which entails that intersecting the wave function at various amplitudes gives a set of varying fractal dimensions depending on the amplitude. A natural question occurs: is it possible to unveil some universality, perhaps based on RMT, for the *ergodic-to-nonergodic transition* itself for a broad range of systems? It should be stressed once more that such RMT-based descriptions are purely phenomenological, and that these do not capture e.g. the dimension-dependence of disorder-induced localization in the genuine Anderson model. Indeed, RMT-based approaches would capture generic features of ergodic-to-nonergodic transitions which includes, but is not limited to, the Anderson transition. This is very much in the same spirit as the well-known fact that Wigner-Dyson statistics describe the energy level statistics of generic quantum chaotic systems, regardless of the microscopic details of these systems.

There were several proposals for an RMT-description of ergodic-to-nonergodic transitions [9], [10], [11], [12], [13], [14], [15], [16]. Since Anderson transition occurs in real space, the RME symmetry should be broken in some way: this is a general feature required for the RMT to describe the transition. One obvious class of RMT's should therefore has a *manifestly* broken symmetry. A notable example of these theories are the banded, non-invariant RMT's [8].

The probability distribution $P(H) \sim \exp(-\sum_{i,j} |H_{ij}|^2/A_{ij})$ is defined by the variance matrix $A_{ij} \sim [1 + (i-j)^2/B^2]^{-1}$, which is clearly non-invariant with respect to the unitary transformations of the form $H \rightarrow UHU^\dagger$. It was explicitly demonstrated that this ensembles describes an intermediate statistics and the multifractal wave functions [17].

However, one can also have intermediate statistics in ensembles where the symmetry is not explicitly broken, i.e. for which the measure is invariant with respect to the transformations from the corresponding group. We focus on these ensembles here. Generically speaking, one can classify ensembles according to the asymptotic behaviour of the confining potential. Let us consider a power-law asymptotic scaling, $V(h) \sim |h|^\alpha$ for $|h| \gg 1$. If the exponent $\alpha$ satisfies $\alpha > 1$, we talk about *steep confinement*. When on the other hand $\alpha < 1$, we deal with a *weakly confined* Random Matrix Ensemble. A particular weakly confined RME may be obtained from the generic one by a limiting procedure. Consider a potential of the form $V_\alpha(h) = \gamma^{-1}h^{-2}(|h|^\alpha - 1)^2$ for large $|h|$. In the limit $\alpha \rightarrow 0$ at fixed $h$, we find the following confining potential

$$V(\mathbf{H}) = \frac{1}{\gamma} \log^2 \mathbf{H}, \quad |\mathbf{H}| \gg 1 , \tag{1}$$

which shall be called *log-Gaussian* critical RME (or a $\log^2$-RME) [18]. It was realized that several classes of *invariant* RMT's, such as (1) exhibit intermediate statistics in terms of eigenvalues and *multifractal* behavior in terms of statistics of its eigenfunctions. Remarkably, both the spectral statistics and eigenvector multifractality at the mobility edge were found to match the matrix ensemble prediction at the exact same value of $q$ [19]. This behavior is somehow reminiscent of the spontaneous symmetry breaking conjectured in [19], [20].

The intermediate RME exists in a 'circular' guise, i.e. where the matrices under consideration are unitary instead of Hermitian, so that its eigenvalues lie on the complex unit circle. In this case, the potential is given by

$$V(x) = \log \left[ \prod_{j=1}^{\infty} (1 + q^{j-1/2}x)(1 + q^{j-1/2}x^{-1}) \right] , \tag{2}$$

which, upon exponentiating, is proportional to the third Jacobi theta function. Again, due to the fact that certain expressions are more tractable in the circular case, we focus on this representation.

## 1.3 Connection to topological field and string theories

The intermediate RME described above was also found in a completely different context, namely, as a matrix model of $U(N)$ Chern-Simons theory $S^3$ [21]. Chern-Simons is a topological theory, indeed, Witten famously showed that its Wilson line expectation values are given by knot- and link invariants [22]. We suspect that it is not a coincidence that the matrix model of a topological theory has intermediate statistics characteristics of ergodic-to-nonergodic transitions. Indeed, the absence of a natural local order parameter in ergodic-to-nonergodic transitions suggest that it is natural to use topological tools for its characterization.

There is, in fact, a relation between strongly Anderson-localized systems and noninteracting topological states [23]. One of the most notable features of topological states of matter is the existence of propagating edge states, which are robust with respect to the application of arbitrarily strong perturbations at the boundary that break translational symmetry (e.g. disorder). It is quite remarkable that these gapless, extended states persist in systems with strong disorder, as this means that such states are robust against Anderson localization. As such, the problem of

classifying all *noninteracting* topological insulators in $d$ spatial bulk dimensions is equivalent to a classification problem of Anderson localization at the $(d-1)$-dimensional boundary. Indeed, the 10-fold classification scheme of noninteracting topological insulators [24] is equivalent to the Altland-Zirnbauer classification of (*noninteracting*) Anderson insulators [25]. This correspondence however does not describe transition from ergodic to nonergodic phases. This begs the question: *can the nonergodic phases and ergodic-to-nonergodic phase transitions be generally related to certain* **interacting** *topological states of matter?*

Indeed, $U(N)$ Chern-Simons theory is such an interacting topological system which describes ergodic-to-nonergodic transitions. We conjecture that it is representative of a broader correspondence, and that the appropriate tools for the description of ergodic-to-nonergodic transitions are available in the topological part of the string theory. This provides a potential new bridge (apart from AdS/CMT duality) between string theory and quantum many-body theory, from which a fruitful exchange of ideas can arise. This is the main motivation of the present work.

To further substantiate our conjecture, we note that close inspection of matrix model potentials which appeared in the context of topological strings (see e.g. [26], [27], [28]) shows that all of them belong to the class of weak confinement potentials, as far as the authors are aware. As described above, weak confinement is a signature of intermediate statistics. On the other hand, it appears that many, if not all, of the known intermediate invariant one-matrix models that appeared in the condensed matter literature and which exhibit a multifractal spectrum are described by some of the variants of *topological string theory*. In the simplest case of the Chern-Simons matrix model, the connection to string theory arises from the finding due to Witten [29] that a $U(N)$ Chern-Simons theory on $S^3$ describes open topological strings on the co-tangent space $T^*S^3$, in the presence of $N$ $D$-branes wrapping $S^3$. Later, Gopakumar and Vafa [30], [31] found that these models correspond to closed topological strings on other spaces, called conifolds. This correspondence was named geometrical transition between a so-called A and B models and is one of the manifestations of the *gauge-gravity duality* (see [32] for an extensive review). In the $N \to \infty$ limit, which we focus on here, $U(N)$ Chern-Simons theory on $S^3$ undergoes a so-called *crystal melting transition* [33], which is related to topological strings on certain Calabi-Yau manifolds [34]. We conjecture that matrix models with a similar origin in topological string theory, such as those of $U(N)$ Chern-Simons theories on general lens spaces or or ABJM theory, also exhibit intermediate statistics.

## 1.4  Summary of main results

To clarify the connection between intermediate RME and topological string theory, we calculate the asymptotic SFF for the Chern-Simons matrix model. The SFF is one of the central objects in RMT, it has clear features which differentiate between ergodic and nonergodic behaviors. While our original motivation was the intermediate Chern-Simons matrix model mentioned above, the techniques we apply have far broader applicability. In particular, they can be applied to any matrix model with unitary matrices of infinite order and weight function satisfying the assumptions of Szegö's limit theorem [35]. For this reason, we treat both the general and the specific cases, so that certain sections may be skipped depending on the particular interests of the reader.

- **Spectral Form Factor**

  To calculate the SFF, we express it as a sum over weighted unitary integrals with the insertion of Schur polynomials. These integrals take the form of certain Toeplitz minors [36], [37], [38], [39]. We assume we can write the weight function as $f(z) = E(x;z)E(x;z^{-1})$ or $f(z) = H(x;z)H(x;z^{-1})$, where $E(x;z)$ $(H(x;z))$ is the generating function of ele-

mentary (homogeneous) symmetric polynomials defined in terms of a set of variables $x = (x_1, x_2, \dots)$. We find that the SFF is then given by

$$\frac{1}{N}\left\langle |\mathrm{tr}U^n|^2 \right\rangle = \begin{cases} N^{-1}\left[ n + p_n(x)^2 \right] & , \quad n/N \leqslant 1 , \\ 1 & , \quad n/N \geqslant 1 , \end{cases} \tag{3}$$

where $p_n(x)$ are power sum polynomials in terms of $x$. SFF's are typically characterized by what has been termed a *dip-ramp-plateau* shape, see e.g. [40], [41], [42]. We find that the *dip* arises from the *disconnected* SFF, i.e. $\left\langle \mathrm{tr}U^n \right\rangle^2 = p_n(x)^2$. The factor $n$ which saturates at $n/N = 1$ gives the *ramp* and *plateau*; this contribution arises from the *connected* SFF.

- **Trace identities**

  As an auxiliary result to the calculation of the SFF, it is easy to show that, for $m, n \in \mathbb{Z}^+$,

  $$\left\langle \mathrm{tr}U^m \mathrm{tr}U^{-n} \right\rangle = m\delta_{mn} + \left\langle \mathrm{tr}U^m \right\rangle\left\langle \mathrm{tr}U^{-n} \right\rangle . \tag{4}$$

  Further, for a partition $\lambda$ satisfying $\lambda_1 + \lambda_1^t - 1 < n$ for some $n \in \mathbb{Z}^+$, we have,

  $$\left\langle \mathrm{tr}_\lambda U \mathrm{tr}U^{-n} \right\rangle = \left\langle \mathrm{tr}_\lambda U \right\rangle\left\langle \mathrm{tr}U^{-n} \right\rangle . \tag{5}$$

  Consider instead the case where $\lambda$ satisfies $\lambda_1 + \lambda_1^t - 1 < n$, and define $m := \lambda_1 + \lambda_1^t - 1 - n$. Then, if $m \leqslant \lambda_1 - \lambda_2$ and $m \leqslant \lambda_1^t - \lambda_2^t$, (5) holds as well.

- **Dualities**

  It is easy to see that, upon replacing $E(x; z)$ by $H(x; z)$, we find exactly the same SFF. Indeed, for any set of variables $x$ for which $(p_n(x))^2$ gives the same value for all $n$, $f(z) = E(x; z)E(x; z^{-1})$ and $f = H(x; z)H(x; z^{-1})$ gives the same SFF. We suspect that this is an example of a larger class of dualities between various intermediate RME's.

- **Application to Chern-Simons RME**

  We apply these results to the matrix model with weight function given by the third Jacobi theta function,

  $$f(z) = \sum_{n \in \mathbb{Z}} q^{n^2/2} z^n = (q; q)_\infty \prod_{k=1}^\infty (1 + q^{k-1/2}z)(1 + q^{k-1/2}z^{-1}) , \quad 0 < |q| < 1 . \tag{6}$$

  This is the matrix model described above, which was introduced in [13] as a phenomenological model of intermediate statistics, and in [43] as a matrix model of $U(N)$ Chern-Simons theory on $S^3$. In the latter context, the SFF is given by a topological invariant, specifically, the *HOMFLY invariant* [44], of $(2n, 2)$-torus links with one component in the fundamental and the other in the antifundamental representation. As far as the authors are aware, these invariants have heretofore not appeared in the literature. As for all matrix models considered here, the SFF is given by a linear ramp which saturates at a plateau, plus a disconnected contribution. Since the SFF corresponds to a $(2n, 2)$-torus link, it follows that the disconnected contribution is the product of two $(n, 1)$-torus knots. Calculating the invariant of an $(n, 1)$-torus knots for general $N$, we find that it is given by the $q^n$-deformation of $N$, which simplifies even further upon implementing the limit $N \to \infty$. We thus find the following expression for the SFF

  $$\frac{1}{N}\left\langle |\mathrm{tr}U^n|^2 \right\rangle = \begin{cases} N^{-1}\left[ n + (q^{-n/2} - q^{n/2})^{-2} \right] & , \quad n/N \leqslant 1 , \\ 1 & , \quad n/N \geqslant 1 . \end{cases} \tag{7}$$

We plot this below for $q = 0.9^k$, $k = 1, \ldots, 9$, where we add lines at $x + (q^x + q^{-x} - 2)^{-1}$ for continuous $x$ as a guide to the eye. The trace identities in (4) and (5) of course apply to the Chern-Simons matrix model as well, where the latter entails that one can 'unlink' an $(n, 1)$-torus knot in the fundamental representation and an unknot in representation $\lambda$.

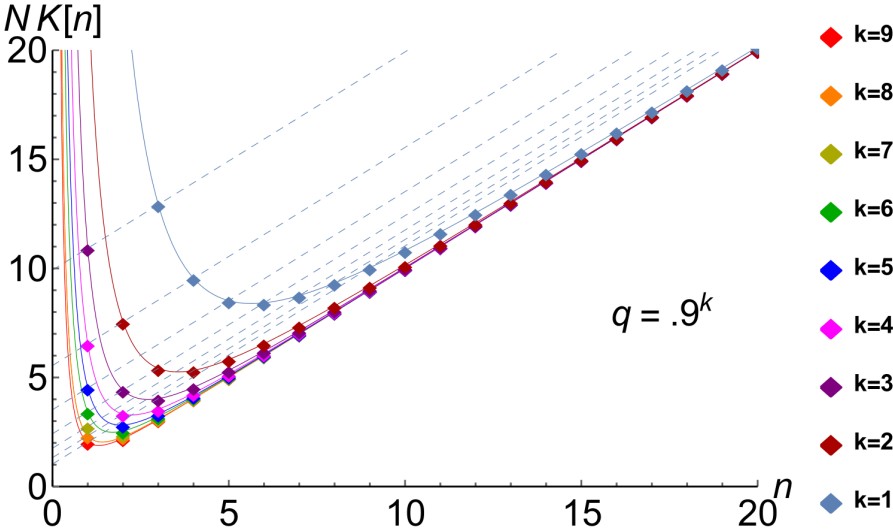

Figure 1: The SFF given in (104) plotted for $n = 1, 2, \ldots, 20$, with $q = 0, 9^k$, $k = 1 \ldots, 9$. The continuous lines are added to guide the eye. For $q$ farther from 0, the disconnected contribution becomes larger, so that the dip is more pronounced and the SFF displays greater deviations from a simple linear ramp. There are dashed lines which indicate $kn = $ constant, in particular $kn = 1, \ldots, 9$. From (91), it follows that lines with $kn = $ constant lie at 45 degrees for *any* SFF calculated here, i.e. any SFF given by (89). Note that these SFF's saturate at a plateau at $n/N = 1$, which is, of course, not indicated in this plot.

## 1.5 Outline of the paper

This paper organized as follows. In section 2, we set up the general framework of random matrix ensembles and introduce important objects, including the SFF. In section 3, we treat $U(N)$ Chern-Simons theory on $S^3$ and its expression as a matrix model, after which we consider the expression of knot and link invariants as matrix integrals. In section 4, we review the computation of such matrix integrals using their expression in terms of Toeplitz minors. These Toeplitz minors, in turn, are given by symmetric polynomials in terms of variables determined by the weight function. We then express the assumptions of Szegö's theorem as requirements on these symmetric polynomials, in particular the power sum polynomials. Further, we find in this section that, although the expression in this work are generally valid for $N \to \infty$, in certain cases they are valid for finite $N$ as well.

In section 5, we set out to compute the SFF using the techniques outlined in the previous sections. Using fundamental relations in the theory of symmetric polynomials, we derive the results for general weight function outlined in the previous subsection. The specific case of the SFF of the Chern-Simons matrix model is worked out in section 5.2. We then consider the broader implications of these calculations in the concluding remarks. In the appendices, the reader can find more details about $q$-deformations and symmetric polynomials, with special attention given to Schur polynomials.

## 2 Random matrix theory

We will consider random matrix ensembles, which have partition functions in the form of a matrix integral,

$$\int dM P(M) \, . \tag{8}$$

Here, $P(M)$ is the probability density function associated to $M$. Consider first the case where the matrices $M$ are Hermitian, so that they can be diagonalized by a unitary transformation. Integrating over $U(N)$ leads to an eigenvalue expression of the form [4]

$$Z = C_N \int \prod_{i=1}^{N} \frac{dx_i}{2\pi} f(x_i) \prod_{i<j} (x_i - x_j)^2 \, , \tag{9}$$

where $C_N$ is some multiplicative constant and $f(x)$ is called the *weight function*. Choosing

$$P(M) \propto \exp(-\alpha \mathrm{tr} M^2) \, , \tag{10}$$

where $\alpha$ is some positive numerical constant, leads to the familiar Gaussian unitary ensemble (GUE) with weight function $f(x) = \exp(-\alpha x^2)$. This ensemble is characterized by fully extended eigenvectors and strong eigenvalue repulsion, which we will collectively refer to as Wigner-Dyson statistics. It was conjectured in the 1980's [45], [46], [47] that the eigenvalues of quantum systems whose classical counterpart is chaotic exhibit Wigner-Dyson statistics (after an unfolding procedure, which is to say, a rescaling of the energies such that the average inter-energy spacing equals unity). This conjecture has been so extensively corroborated that Wigner-Dyson statistics are nowadays seen almost as a definition of quantum chaos.

We will also consider ensembles whose elements are themselves unitary matrices. Historically, the first example of such an ensemble is the CUE introduced by Dyson [3], which is mentioned in the introduction. Being unitary, the eigenvalues of these matrices are distributed across the complex unit circle. Such unitary ensembles have a partition function of the form

$$Z = \tilde{C}_N \int \prod_{i=1}^{N} \frac{d\phi_i}{2\pi} f(\phi_i) \prod_{i<j} |e^{-i\phi_i} - e^{-i\phi_j}|^2 \, , \tag{11}$$

where we denote the matrices under consideration by $U$. For $f(x_i)$=constant, (11) reduces to Dyson's circular unitary ensemble (CUE). in the limit $N \to \infty$, the CUE and GUE exhibit the same bulk statistics after unfolding, i.e. the CUE also described systems whose classical counterpart is chaotic [4], [48].

While the Wigner-Dyson ensembles described above provide excellent phenomenological descriptions of quantum chaotic systems, they naturally fail to describe systems with intermediate spectral statistics. An example of such a system consists of disordered electrons at the mobility edge of the Anderson localization transition [49], [8]. Muttalib and collaborators introduced a family of random matrix ensembles [13] depending on some parameter $0 \leqslant q \leqslant 1$. This matrix ensemble appears in two guises, analogous to GUE and CUE. In case the matrices under consideration are Hermitian, the weight function is of the following "log-squared" form

$$f(x) \propto \exp\left(-\frac{1}{2g_s} \log^2 x\right) \, , \quad |x| \gg 1 \, . \tag{12}$$

In the expression above, we define $q =: e^{-g_s}$, where $g_s$ is the string coupling constant in the manifestation of Chern-Simons theory as a topological string theory on the cotangent space.

The domain of $f(x)$ in (12) is the positive real line. In case the matrices we consider are themselves unitary, the weight function is given by

$$f(e^{i\phi}) = \Theta_3(e^{i\phi}; q) = \sum_n q^{n^2/2} e^{in\phi} . \tag{13}$$

That is, the weight function is given by Jacobi's third theta function, which is defined on the complex unit circle. One immediately sees that the matrix model with this weight function reduces to the CUE as we take $q \to 0$. We will consider this limit in more detail below. On the other hand, it was shown in [13] that the Hermitian version of the matrix model with weight function given in a certain limit by (12), after unfolding, displays GUE statistics as one takes $q \to 1$. We intend to treat the Hermitian version of our results in general and of weight function (12) in particular in a future work.

## 2.1  Density of states and spectral form factor

An important object in random matrix theory is the density of states, given by

$$\rho(\phi) = \frac{1}{N} \sum_{i=1}^{N} \delta(\phi - \phi_i) = \frac{1}{2\pi N} \sum_{i=1}^{N} \sum_{n \in \mathbb{Z}} e^{in(\phi - \phi_i)} = \frac{1}{2\pi N} \sum_{n \in \mathbb{Z}} \mathrm{tr} U^n e^{in\phi} , \tag{14}$$

where we used the fact that

$$\mathrm{tr} U^n = \sum_{i=1}^{N} e^{-in\phi_i} . \tag{15}$$

The density of states, averaged over the matrix ensemble, gives the probability of finding an eigenvalue at $\phi$. From these level densities, we can construct the $n$-point density correlation functions for $n = 2,\ldots$ and various related quantities. An important example thereof which is often used to characterize the eigenvalue statistics of various ensembles is the SFF, which is the Fourier transform of the two-point level density correlation function [4]. The two-point correlation function is given by,

$$\langle \rho(\theta)\rho(\phi) \rangle = \frac{1}{N^2} \sum_{k,l \in \mathbb{Z}} \langle \mathrm{tr} U^k \mathrm{tr} U^l \rangle e^{ik\theta + il\phi} - 1 . \tag{16}$$

The SFF is then defined as the expansion coefficients of $e^{in(\theta - \phi)}$, $n \in \mathbb{Z}^+$, rescaled by a factor $N$, [4], [48],

$$K(n) = \frac{1}{N} \langle |\mathrm{tr} U^n|^2 \rangle . \tag{17}$$

The choice of normalization is made so that the CUE SFF saturates at unity. For future convenience, we also define the connected part of the SFF

$$K(n)_c = K(n) - \frac{1}{N} \langle \mathrm{tr} U^n \rangle^2 . \tag{18}$$

For the CUE and GUE, the SFF is characterized by a linear ramp which saturates at $n = N$. For intermediate statistics, $K(n)$ displays deviations from this behavior, which can be seen in figure 1 and which will be further detailed below.

# 3 Chern-Simons matrix model and knot/link invariants

## 3.1 Knot operator formalism

We review the construction of Chern-Simons partition functions and knot invariants using Heegaard splitting [22] and knot operators [50]. Heegaard splitting provides a way to calculate the Chern-Simons partition functions of certain three-manifolds, which we denote by $M$. We construct $M$ by taking two separate three-manifolds $M_1$ and $M_2$ which share a common boundary $\Sigma$, i.e. $\partial M_1 \simeq \Sigma \simeq \partial M_2$. $M$ is then constructed by acting on the common boundary $\Sigma$ with some homeomorphism $f$ and then gluing $M_1$ and $M_2$ together, which we write as

$$M = M_1 \bigcup_f M_2 \, . \tag{19}$$

In this construction, we take the boundaries of $M_1$ and $M_2$ to have opposite orientation, so that $M$ is a closed manifold. Writing the Hilbert space of $\Sigma$ as $\mathcal{H}(\Sigma)$ and its conjugate as $\mathcal{H}^*(\Sigma)$, performing the path integral over $M_1$ gives a state $\left|\Psi_{M_1}\right\rangle \in \mathcal{H}(\Sigma)$, whereas performing the path integral over $M_2$ to find a state $\left\langle\Psi_{M_2}\right|$ in the conjugate Hilbert space $\mathcal{H}^*(\Sigma)$ due to the fact that the boundaries of $M_1$ and $M_2$ have opposite orientation. The homeomorphism $f$ induces a map $U_f$ on $\mathcal{H}(\Sigma)$ whose action we denote by

$$U_f : \mathcal{H}(\Sigma) \rightarrow \mathcal{H}(\Sigma) \, . \tag{20}$$

The partition function is then given by

$$Z(M) = \left\langle\Psi_{M_1}\middle|U_f\middle|\Psi_{M_2}\right\rangle \, . \tag{21}$$

In a seminal paper [22], Witten found that $\mathcal{H}(\Sigma)$ is given by the space of conformal blocks of the corresponding Wess-Zumino-Novikov-Witten (WZNW) model on $\Sigma$ at level $k$. In case there are no marked points on $\Sigma$ where Wilson lines are cut, i.e. if all Wilson lines can be embedded on $\Sigma$, $\mathcal{H}(\Sigma)$ is given by the characters of the WZNW model on $\Sigma$. We will be considering only the latter case.

A relatively simple example of a Heegaard splitting is given by the division of $S^3$ into two three-balls that share a boundary $\Sigma = S^2$. The only knot that can be embedded on $S^2$ is the unknot, which is the trivial example of an unknotted circle. We therefore do not consider this example any further. Let us instead consider the case where $M_1$ and $M_2$ are given by solid tori $S^1 \times D^2$ which share a boundary torus $\partial M_1 = S^1 \times S^1 = \partial M_2$. The manifolds which can be constructed via such a Heegaard splitting on a torus are known as lens spaces [51]. The simplest example of a lens space is found by taking $f$ to be the identity map. In this case, we glue the two copies of $D^2$ along their boundaries to form $S^2$, so that the resulting space is given by $S^2 \times S^1$. We normalize the Chern-Simons partition function for $S^2 \times S^1$ to unity. Let us consider an example where we act on $T^2$ with a nontrivial homeomorphism. The group of homeomorphisms of $T^2$ is given by $SL(2;\mathbb{Z})$, which consists of matrices of the form

$$\begin{pmatrix} a & b \\ c & d \end{pmatrix} \, , \quad ad - bc = 1 \, , \quad a, b, c, d \in \mathbb{Z} \, . \tag{22}$$

$SL(2;\mathbb{Z})$ is generated by the modular $S$ and $T$-transformations. Representing the 1-cycles of the torus by basis vectors $\begin{pmatrix} 1 \\ 0 \end{pmatrix}$ and $\begin{pmatrix} 0 \\ 1 \end{pmatrix}$, the $S$ and $T$-transformations can be written as

$$S = \begin{pmatrix} 0 & -1 \\ 1 & 0 \end{pmatrix} \, , \quad T = \begin{pmatrix} 1 & 1 \\ 0 & 1 \end{pmatrix} \, . \tag{23}$$

That is, $S$ interchanges the 1-cycles and reverses the orientation of the torus, while $T$ cuts open the torus along a 1-cycles to form a cylinder, twists one end of the cylinder by $2\pi$, and glues the two ends of the cylinder back together. Consider the case where we glue two solid tori $M_{1,2}$ along their boundaries after acting with an $S$-transformation. Since $S$-transformations exchange the 1-cycles on the torus, the contractible cycle of $M_1$ is glued to the non-contractible cycle of $M_2$ and vice versa. We thus find a closed three-manifold with no non-contractible cycles which, from the Poincaré conjecture, is homeomorphic to $S^3$.

The construction of torus knots is analogous to the construction of lens spaces in the sense that, if we insert a Wilson line corresponding to an unknot on the boundary torus, we can act with arbitrary $SL(2;\mathbb{Z})$ transformation on the torus which turns the unknot into a non-trivial torus knot. Let us denote the torus knot operators, to be defined more precisely below, by $\mathcal{W}^{(p,q)}_\lambda$, where $\lambda$ labels the irreducible representation of the Wilson line and $p$ and $q$ are integers which count the winding of the knot around non-contractible and contractible cycle of the torus, respectively. Note that $p$ and $q$ are coprime for torus knots, whereas for $p$ and $q$ not coprime we would get a torus link, which is a generalization of a torus knot with more than one component (i.e. more than one knotted piece of string). The number of components of a torus link equals the greatest common divisor of $p$ and $q$. From the definition of the $S$ and $T$-transformations, it is clear that they act on torus knot as follows

$$S^{-1}\mathcal{W}^{(p,q)}S = \mathcal{W}^{(q,-p)} \,,$$
$$T^{-1}\mathcal{W}^{(p,q)}T = \mathcal{W}^{(p,q+p)} \,.$$

For example, if we insert an unknot around the non-contractible cycle of the torus and act $n$ times with the $T$-transformation, we get a knot which still winds around the non-contractible cycle once but which now also winds around the contractible cycle $n$ times. Note that this is topologically still an unknot; the additional winding around the contractible cycle only gives rise to a multiplicative framing factor. Similar knots will play an important role in the comparison with random matrix theory, to be outlined below.

It is easy to see that modular transformations map the set of torus knots into itself, as these transformations do not change the number of components. Indeed, for any pair of coprime integers $(p,q)$, one can easily see that $(p,q+p)$ are also coprime, so that the number of components is unchanged under modular transformations. Further, due to Bézout's lemma [52], there is an $SL(2;\mathbb{Z})$-transformation corresponding to any pair of coprime integers, so that we can construct any torus knot by acting on an unknot with an $SL(2;\mathbb{Z})$-transformation.

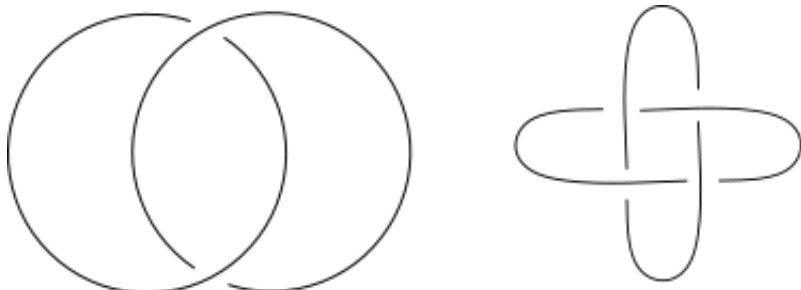

Figure 2: Two examples of $(2n, 2)$-torus links. The Hopf link, on the left, is the $(2, 2)$-torus link. On the right, we have the $(4, 2)$-torus link.

The explicit form for the knot operators mentioned above was found by Labastida, Llatas, and Ramallo [50], using the relation to WZNW-models previously found by Witten [22]. Let us summarize the salient points of the knot operator formalism. As mentioned above, $\mathcal{H}(\Sigma)$ is given by the conformal blocks of the corresponding WZNW-model on $\Sigma$ with group $G$ at level $k$. In the case of $\Sigma = T^2$ without marked points, which we will be considering henceforth, $\mathcal{H}(\Sigma)$ consists of the characters of integrable representations of the corresponding WZNW-model. We denote the set of fundamental weights by $\{v_i\}$ and Weyl vector by $\rho = \sum_i v_i$. A representation with highest weight $\Lambda$ is integrable if $p := \rho + \Lambda = \sum_i p_i v_i$ is in the fundamental Weyl chamber, that is,

$$\sum_i p_i < k + y \ , \quad p_i > 0 \ , \ \forall \, i \, , \tag{24}$$

where $y$ is the dual Coxeter number of $G$, which equals $N$ for $G = U(N)$ and $N - 1$ for $G = SU(N)$. Remember that an irrep with highest weight $\Lambda = \sum_i \Lambda_i v_i$ corresponds to a Young tableau where the length of the $i^{\text{th}}$ row is given by

$$\Lambda_i + \Lambda_{i+1} + \cdots + \Lambda_I \, , \tag{25}$$

where $I$ equals $N$ in the case of $U(N)$ and $N - 1$ in the case of $SU(N)$. See appendix B or e.g. section 13.3.2 of [53] for more background information on partitions and their role in representation theory. From now on we will take $G = U(N)$ so that $y = N$. We will denote ket states corresponding to $p$ by $|p\rangle$, which can be chosen in such a way that they form an orthonormal basis. The vacuum state, that is, the state without any Wilson line inserted, is given by $|\rho\rangle =: |0\rangle$. If we act with a knot operator corresponding to an unknot in representation corresponding to $\Lambda$, the result is [50]

$$\mathcal{W}_\Lambda^{(1,0)} |\rho\rangle = |\rho + \Lambda\rangle = |p\rangle \, . \tag{26}$$

The only further ingredient we need are the explicit expressions for the Hilbert space operators induced by the modular transformations. We simply state these here, further details may be found in [50]

$$T_{pp'} = \delta_{p,p'} e^{2\pi i(h_p - c/24)} \, ,$$
$$S_{pp'} = \frac{i^{N(N-1)/2}}{N^{N/2}} \left( \frac{N}{k+N} \right)^{\frac{N-1}{2}} \sum_{w \in W} \epsilon(w) \exp \left( \frac{-2\pi i p \cdot w(p')}{k+N} \right) \, . \tag{27}$$

In the above expressions, $W$ is the Weyl group, $\epsilon(w)$ is the signature of Weyl reflection $w$, $c$ is the central charge of the WZNW-model, and $h_p$ is the conformal weight of the primary field corresponding to $p$, which is given by

$$h_p = \frac{p^2 - \rho^2}{2(k+y)} \, . \tag{28}$$

## 3.2 Chern-Simons matrix model

Let us consider how the matrix model description of Chern-Simons theory arises. As explained above, $S^3$ can be constructed via a Heegaard splitting along a torus on which we act with an $S$-transformation. We thus find that the Chern-Simons partition function on $S^3$ is given by

$$Z(S^3) = \langle 0|S|0\rangle = S_{00} \, . \tag{29}$$

We plug in the expression for $S_{00}$ from equation (27) and use Weyl's denominator formula,

$$\sum_{w \in W} \epsilon(w) e^{w(p)} = \prod_{\alpha > 0} 2\sinh(\alpha/2) \, , \tag{30}$$

where $\alpha$ are the positive roots of $U(N)$. Expressing the roots of $U(N)$ in terms of Dynkin coordinates $x_i$, we find

$$Z(S^3) = \frac{e^{-\frac{g_s}{12}N(N^2-1)}}{N!} \int \frac{dx_i}{2\pi} \prod_{i=1}^{N} e^{-x_i^2/2g_s} \prod_{i<j} \left( 2\sinh\frac{x_i - x_j}{2} \right)^2 . \tag{31}$$

Lastly, we define a new set of variables $y_i := e^{Ng_s + x_i}$, in which the partition function is given by [43]

$$Z(S^3) = \frac{e^{-\left(7N^3 g_s/12 + N^2 g_s/2 - N g_s/24\right)}}{N!} \int_0^\infty \prod_{i=1}^{N} \frac{dy_i}{2\pi} \prod_{i<j} (y_i - y_j)^2 \exp\left( -\frac{1}{2g_s} \sum_i \log^2(y_i) \right) . \tag{32}$$

Alternatively, we can use the following expression

$$q^{n^2/2} = \int_0^{2\pi} \frac{d\phi}{2\pi} \Theta_3(e^{i\phi}; q) e^{in\phi} , \tag{33}$$

where we repeat the definition of the third Jacobi Theta function

$$\Theta_3(e^{i\phi}; q) = \sum_{n \in \mathbb{Z}} q^{n^2/2} e^{in\phi} . \tag{34}$$

This gives

$$\begin{aligned}
\sum_{w \in W} \epsilon(w) q^{\frac{1}{2}(w(\rho)-\rho)^2} &= \frac{1}{|W|} \sum_{w,w' \in W} \epsilon(w)\epsilon(w') q^{\frac{1}{2}(w(\rho)-w(\rho'))^2} \\
&= \frac{1}{|W|} \int \prod_{i=1}^{N} \frac{d\phi_i}{2\pi} \Theta_3(e^{i\phi_i}; q) \sum_{w,w' \in W} \epsilon(w)\epsilon(w') q^{i(w(\rho)-w(\rho')\cdot\theta} ,
\end{aligned} \tag{35}$$

where we added another summation over the Weyl group in the first equality and applied (33) in the second. Lastly, the Weyl group $W$ is isomorphic to the symmetric group $S_N$ so that $|W| = N!$. Using the Weyl denominator formula again leads to [34] [54].

$$Z = \frac{1}{N!} \int_0^{2\pi} \prod_{i=1}^{N} \frac{d\phi_i}{2\pi} \Theta_3(e^{i\phi_j}; q) \prod_{j<k} |e^{i\phi_j} - e^{i\phi_k}|^2 . \tag{36}$$

Note that (32) and (36) correspond precisely to the matrix ensemble introduced by [13], given in (12) and (13), respectively. Further, using the Jacobi triple product formula, $\Theta_3$ can be written as a specialization of $E(x; z)$, the generating function of the elementary symmetric polynomials. We can also replace $E(x; z)$ by $H(x; z)$ at the cost of transposing all representations involved in the calculation, this amounts to replacing $\Theta_3(z; q)$ by $\frac{1}{\Theta_3(-z; q)}$. Since the SFF is invariant under transposition of all representations (see e.g. (78)), the calculation of the SFF done below is also valid for the case where the weight function of of the form $\frac{1}{\Theta_3}$. Indeed, the above argument applies to any specialization i.e. to any choice of variables $x_i$. We will therefore use $E(x; z)$ and $H(x; z)$ interchangeably in the computations below.

## 3.3 Computing torus knot and link invariants in the Chern-Simons matrix model

We now consider knot and link invariants and their computation in the Chern-Simons matrix model. First, we treat the multiplication properties of knot operators. If we take $\mathcal{W}_\lambda^{\mathcal{K}}$ to be a knot operator corresponding to a knot $\mathcal{K}$ in representation $\lambda$, we can write

$$\mathcal{W}_\lambda^{\mathcal{K}} \mathcal{W}_\mu^{\mathcal{K}} = \sum_\nu N_{\lambda\mu}^\nu \mathcal{W}_\nu^{\mathcal{K}} . \tag{37}$$

The coefficients $N_{\lambda\mu}^{\gamma}$ in (37) are the fusion coefficients of the WZNW-model. When both $k$ and $N$ are much larger than any of the representations under consideration, $N_{R_1 R_2}^{R}$ are given by Littlewood-Richardson coefficients. This allows us to construct the invariants of torus links. We label a torus link by $P, Q \in \mathbb{Z}$, where the number of components is given by $S = \gcd(P, Q)$ and the representations are labelled by $j \in \{1, \ldots, S\}$. These links are given by [50], [55], [56]

$$\prod_{j=1}^{S} \mathcal{W}_{\lambda_j}^{P/S,Q/S} = \sum_{\mu} N_{\lambda_1,\ldots,\lambda_S}^{\mu} \mathcal{W}_{\mu}^{P/S,Q/S} , \tag{38}$$

where $N_{\lambda_1,\ldots,\lambda_S}^{\mu}$ are generalized Littlewood-Richardson coefficients appearing in the product of representations $\lambda_1 \otimes \cdots \otimes \lambda_S$.

We now outline the computation of torus knot and link invariants using the matrix model for $U(N)$ Chern-Simons on $S^3$. The simplest knot, the unknot, is given by the ensemble average of the matrix trace in the corresponding representation [57]. That is,

$$W_{\lambda} := \left\langle \mathcal{W}_{\lambda}^{(1,0)} \right\rangle = \langle \mathrm{tr}_{\lambda} U \rangle . \tag{39}$$

If we diagonalize a matrix $U$ to give $\mathrm{diag}(d_1, d_2, \ldots, d_N)$, it is well known that

$$\mathrm{tr}_{\lambda} U = s_{\lambda}(d_1, d_2, \ldots, d_N) = s_{\lambda}(d) , \tag{40}$$

where $s_{\lambda}(d)$ is the *Schur polynomial* corresponding to representation $\lambda$ in terms of variables $d_i$. The reader can consult appendix C or the book by Macdonald [58] or Stanley [59] for more information on Schur polynomials. In the remainder of this work, we will often write traces without specified representations, in which case the trace is understood to be in the fundamental representation.

In general, we can assign an orientation to a knot or component of a link, which corresponds to a continuous non-zero tangent vector along $\mathcal{K}$. When we project a knot or link into the plane, we can assign a sign $+$ or $-$ to each crossing, as in figure 3.

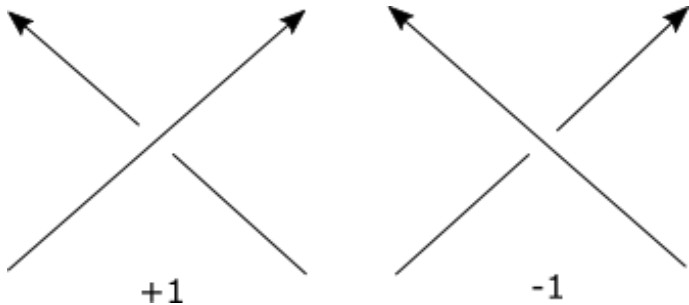

$$+1 \qquad\qquad -1$$

Figure 3: After projecting a knot or link in the plane, crossings are given a sign in the way indicated above.

We denote by $\overline{\lambda}$ the representation conjugate to $\lambda$. We then have [21]

$$\mathrm{tr}_{\lambda} U^{-1} = \mathrm{tr}_{\overline{\lambda}} U , \tag{41}$$

in the language of knot theory, taking $\mathrm{tr}_{\lambda} U$ to $\mathrm{tr}_{\lambda} U^{-1}$ corresponds to inverting the orientation of the component carrying representation $\lambda$. Of course, for the unknot, this does not matter, as

reverting the orientation can be compensated by a simple parity transformation. The same is true for the Hopf link, as overcrossings can be freely changed into undercrossings. To convince oneself of this point, one can assign an orientation to both components of the Hopf link in figure 2, and rotate one component along an axis parallel to the projection plane whilst keeping the other component fixed. For more complicated knots or links, such as the $(4, 2)$-torus link on the right hand side of figure 2, overcrossings can no longer be turned into undercrossings and inverting the orientation of one component will generally lead to a different expectation value.

Let us consider more complicated objects involving integer powers of $U$. Generally, any product of traces of any $GL(N, \mathbb{C})$ matrix $U$,

$$S_\alpha = (\text{tr} U)^{\alpha_1} (\text{tr} U^2)^{\alpha_2} \ldots (\text{tr} U^s)^{\alpha_s} \ , \quad \alpha_i \in \mathbb{Z}^+ \ , \tag{42}$$

can be expanded in characters of $GL(N, \mathbb{C})$, denoted by $\chi_\lambda(U)$, with characters of the symmetric group $S_l$ as expansion coefficients, where $l = \sum_i \alpha_i$ [60]. If $U \in U(N)$, the characters are given by Schur polynomials, see appendix C for more background. We then have

$$\text{tr} U^{\alpha_1} \text{tr} U^{\alpha_2} \ldots \text{tr} U^{\alpha_k} = \sum_R \chi_R(C(\vec{k})) \text{tr}_R U \ , \tag{43}$$

where $\sum_R$ is a sum over all Young tableaux with total number of boxes equal to $l$, and $\chi_R(C(\vec{k}))$ is the character of the symmetric group $S_l$ in representation $R$ evaluated at the conjugacy class of $S_l$ given by cycle lengths $\alpha_1, \alpha_2, \ldots, \alpha_k$. Despite its concise notation, (43) is generally rather difficult to compute due to the sum over partitions of $l$. However, in certain cases the above expression can be calculated. Taking $U \in U(N)$ with eigenvalues $d_i$ and choosing $\alpha_1 = n$ and $\alpha_i = 0$ for $i \neq 1$, we find [60]

$$\text{tr} U^n = \sum_i d_i^n = \sum_\lambda \chi_{(n)}^\lambda s_\lambda(U) = \sum_{r=0}^{n-1} (-1)^r s_{(n-r, 1^r)}(U) \ , \tag{44}$$

where we used the fact that characters of the symmetric group satisfy

$$\chi_{(n)}^\lambda = \begin{cases} (-1)^r \ , & \text{if } \lambda = (n-r, 1^r) \ , \\ 0 \ , & \text{otherwise} \ . \end{cases} \tag{45}$$

In words, (44) states that $\text{tr} U^n$ is given by the sum over hook-shaped irreps with $n$ boxes, which appear with alternating signs. One may recognize from (44) that this is the expression of the $n^{\text{th}}$ power sum polynomial in terms of Schur polynomials. For $n = 4$, one can express (44) in terms of Young diagrams as follows.

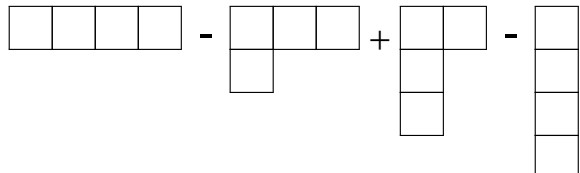

One can show [57], [61], [62] that $\langle \text{tr} U^n \rangle$ gives the invariant of an $(n, 1)$-torus knot [63], which differs from any $(n, m)$-torus knot only by a framing factor. Equation (44) gives an expansion of of $\text{tr} U^n$ in terms of Schur polynomials. Explicit expressions for its expectation value can be found in section 5.2. As noted above, an $(n, 1)$-torus knot is topologically equivalent to an unknot and differs only due to framing [57]. However, terms of the form $\langle \text{tr} U^n \text{tr} U^{-n} \rangle$, such as appear in the SFF, give $(2n, 2)$-torus links, which are not topologically trivial for any $n \in \mathbb{Z} \backslash \{0\}$.

# 4 Matrix integrals and Toeplitz minors

We review the computation of the unitary group integral over Schur polynomials using a method outlined in [38] and [39], which in turn draw from results derived by Bump and Diaconis [36], Tracy and Widom [37], among others. We start from an absolutely integrable function on the unit circle in $\mathbb{C}$,

$$f(e^{i\theta}) = \sum_{k \in \mathbb{Z}} d_k e^{ik\theta} . \tag{46}$$

We will specifically be considering the case where $d_k = d_{-k}$, so that $f(e^{i\theta})$ is real-valued. We further require that $f(e^{i\theta})$ satisfies the assumptions of Szegö's theorem. That is, we write $f(e^{i\theta})$ as

$$f(e^{i\theta}) = \exp\left(\sum_{k \in \mathbb{Z}} c_k e^{ik\theta}\right) , \tag{47}$$

and demand that

$$\sum_{k \in \mathbb{Z}} |c_k| < \infty , \quad \sum_{k \in \mathbb{Z}} |k||c_k|^2 < \infty . \tag{48}$$

From the Fourier coefficients of $f$, we construct a *Toeplitz matrix*, which is a matrix that is constant along its diagonals,

$$T(f) = (d_{j-k})_{j,k \geqslant 1} . \tag{49}$$

We denote by $T_N(f)$ the $N$ by $N$ principal submatrix of $T(f)$, i.e. the matrix obtained from $T(f)$ by taking its first $N$ rows and columns and neglecting the remainder. We will see that various matrix integrals with weight function $f$ can be expressed as minors of $T_N(f)$, that is, as determinants of matrices obtained from $T_N(f)$ by removing a (necessarily equal) number of rows and columns. For a unitary matrix $U$ with eigenvalues $e^{i\theta_1}, e^{i\theta_2}, \ldots$, we write,

$$\tilde{f}(U) = \prod_{k=1}^{N} f(e^{i\theta_k}) . \tag{50}$$

We employ Weyl's integral formula [64] to express the integral of $\tilde{f}(U)$ over $U(N)$ with respect to the de Haar measure as

$$\int \tilde{f}(U)dU = \frac{1}{N!} \int_0^{2\pi} \prod_{j<k} |e^{i\theta_j} - e^{i\theta_k}|^2 \prod_{k=1}^{N} f(e^{i\theta_k}) \frac{d\theta_k}{2\pi} , \tag{51}$$

where the angles satisfy $0 \leqslant \theta_k < 2\pi$. The expression for the Vandermonde determinant in (132) allows us to use an identity due to Andreiéf, sometimes referred to as Heine or Gram identity [65]. Take $g_j$ and $h_j$, $j \in \{1, 2, \ldots, N\}$, to be two sequences of integrable functions on some measure space with measure $\mu$, then

$$\frac{1}{N!} \int \det(g_j(x_k))_{j,k=1}^{N} \det(h_j(x_k)_{j,k=1}^{N} \prod_{k=1}^{N} d\mu(x_k) = \det\left(\int g_j(x)h_j(x)d\mu(x)\right)_{j,k=1}^{N} . \tag{52}$$

Choosing $g_j(e^{-i\theta}) = e^{i(N-j)\theta} = h_j(e^{i\theta})$ and $d\mu(e^{i\theta}) = f(e^{i\theta})\frac{d\theta}{2\pi}$, we find

$$\int \tilde{f}(U)dU = \det(d_{j-k})_{j,k=1}^{N} , \tag{53}$$

where $d_k$ are again the Fourier coefficients of $f$,

$$d_k = \frac{1}{2\pi} \int_0^{2\pi} f(e^{i\theta})e^{ik\theta}d\theta . \tag{54}$$

Now let $\lambda = (\lambda_1, \ldots, \lambda_m)$ and $\mu = (\mu_1, \ldots, \mu_n)$ be partitions of $|\lambda| = \sum_i^{\ell(\lambda)} \lambda_i$ and $|\mu| = \sum_j^{\ell(\mu)} \mu_j$, respectively. Here, $\lambda_i$, $\mu_j \in \mathbb{Z}^+$ and $\ell(.)$ is the length of the partition. Ordering as $\lambda_i \geqslant \lambda_{i+1}$ and similarly for $\mu_j$, these partitions label Young tableaux in the standard way. One then obtains a *Toeplitz minor* $T_N^{\lambda,\mu}(f)$ via the following procedure:

- We start from $T_{N+\kappa}(f)$, where $\kappa = \max\{\lambda_1, \mu_1\}$

- If $\lambda_1 - \mu_1 > 0$, we remove the first $\lambda_1 - \mu_1$ colums from $T_{N+\kappa}(f)$, otherwise we remove $\mu_1 - \lambda_1$ rows.

- We then keep the first row and remove the next $\lambda_1 - \lambda_2$ rows, after which we again keep the first row and remove the next $\lambda_2 - \lambda_3$ rows and so on and so forth.

- We repeat the third step where we replace $\lambda_i$ by $\mu_i$ and where we remove columns instead of rows

Note that the second step ensures that the resulting matrix $T_N^{\lambda,\mu}(f)$ is of order $N$. We write $s_\lambda(U) = s_\lambda(e^{i\theta_1}, e^{i\theta_2}, \ldots)$, where $s_\lambda$ are Schur polynomials, which we review in appendix C. The determinant of $T_N^{\lambda,\mu}(f)$ can then be expressed as [36], [66]

$$
\begin{aligned}
D_N^{\lambda,\mu}(f) := \det T_N^{\lambda,\mu}(f) &= \int_{U(N)} s_\lambda(U^{-1}) s_\mu(U) \tilde{f}(U) dU \\
&= \frac{1}{N!(2\pi)^N} \int_0^{2\pi} s_\lambda(e^{-i\theta_1}, \ldots, e^{-i\theta_N}) s_\mu(e^{i\theta_1}, \ldots, e^{i\theta_N}) \prod_{j=1}^{N} f(e^{i\theta_j}) \prod_{1 \leqslant j < k \leqslant N} |e^{i\theta_j} - e^{i\theta_k}|^2 d\theta_j , \\
&= \det \left( d_{j-\lambda_j - k + \mu_k} \right)_{j,k=1}^{N} .
\end{aligned}
\tag{55}
$$

One can recognize the pattern of striking rows and columns involved in the construction of $T_N^{\lambda,\mu}(f)$, as the index $j$ is shifted to $j - \lambda_j$ and $k$ to $k - \mu_k$. One can easily verify that, for two functions of the form

$$
a(e^{i\theta}) = \sum_{k \leqslant 0} a_k e^{ik\theta} , \quad b(e^{i\theta}) = \sum_{k \geqslant 0} b_k e^{ik\theta} ,
\tag{56}
$$

the associated Toeplitz matrix satisfies

$$
T(ab) = T(a)T(b) .
\tag{57}
$$

Let us therefore write $f(e^{i\theta})$ as follows

$$
f(e^{i\theta}) = H(x; e^{i\theta}) H(y; e^{-i\theta}) ,
\tag{58}
$$

where $H(x; z)$ is the generating function of the homogeneous symmetric polynomials $h_k$ given in (123) and where we assume that $h_k(x)$ and $h_k(y)$ are square-summable, i.e.

$$
\sum_k h_k < \infty .
\tag{59}
$$

Gessel [67] showed that, for $f$ as in (58),

$$
D_N(f) = \sum_{\ell(\nu) \leqslant N} s_\nu(x) s_\nu(y) ,
\tag{60}
$$

where one should note that the sum runs over all partitions $\nu$ with at most $N$ rows. Here, we only consider the case where $y = x \in \mathbb{R}$, but the expressions here easily generalize to

$x \neq y$ and $x, y \in \mathbb{C}$, subject to the assumptions of Szegö's theorem. Equation (60) can then be generalized as [38], [39]

$$\int s_\lambda(U^{-1}) s_\mu(U) \tilde{f}(U) dU = \sum_{\ell(\nu) \leqslant N} s_{\nu/\lambda}(x) s_{\nu/\mu}(x) . \tag{61}$$

In the above expressions, we can replace $H(x; z)$ by $E(x; z)$ if we simultaneously transpose all partitions. Let us therefore consider the Jacobi triple product expansion of the third theta function

$$\sum_{n \in \mathbb{Z}} q^{n^2/2} e^{in\theta} = (q; q)_\infty \prod_{j=1}^\infty (1 + q^{k-1/2} e^{i\theta})(1 + q^{k-1/2} e^{-i\theta})$$
$$= (q; q)_\infty E(x; e^{i\theta}) E(x; e^{-i\theta}) , \tag{62}$$

where we define $x = (q^{1/2}, q^{3/2}, \dots)$ in the last line. Then, $f(e^{i\theta}) = \sqrt{(q; q)_\infty} \, E(x; e^{i\theta}) E(x; e^{-i\theta})$ is the weight function of the Chern-Simons matrix model. This example is treated extensively in [39], more details and proofs can be found there. Using (53) with $d_k = q^{k^2/2}$, we see that the partition function is given by

$$Z_N = \int \tilde{f}(U) dU = \det(q^{(j-k)^2/2})_{j,k=1}^N = q^{\sum_{j=1}^N j^2} \det(q^{-jk})_{j,k=1}^N = \prod_{j=1}^{N-1} (1 - q^j)^{N-j} , \tag{63}$$

which is a well-known result.

## 4.1 Infinite $N$

Let us now take the limit $N \to \infty$. From (61) and the fact that [Chapter I.5, example 26 in [58]]

$$\sum_\nu s_{\nu/\mu}(y) s_{\nu/\lambda}(x) = \sum_\nu s_{\lambda/\nu}(y) s_{\mu/\nu}(x) \sum_\kappa s_\kappa(y) s_\kappa(x) , \tag{64}$$

where the sums run over all partitions, we have [38], [39]

$$W_{\lambda\mu} := \frac{\int s_\lambda(U^{-1}) s_\mu(U) \tilde{f}(U) dU}{\int \tilde{f}(U) dU} = \sum_\nu s_{\lambda/\nu}(x) s_{\mu/\nu}(x) . \tag{65}$$

Taking (65) with $\mu = \varnothing$, we see that calculating the matrix integral of a single trace in some representation (55) is given by the following procedure. The evaluation of the integral amounts to replacing the eigenvalues of the Schur polynomials by the variables $x_i$ in $f(z) = E(x; z) E(x; z^{-1})$ or $f(z) = H(x; z) H(x; z^{-1})$. For $f(e^{i\theta})$ equal to $\Theta_3(e^{i\theta})$ in (62), $W_{\lambda\mu}$ gives the HOMFLY invariant of the Hopf link [38], [39]. We see that is is given by the following expression,

$$W_{\lambda\mu} = \sum_\nu s_{(\lambda/\nu)^t}(q^{1/2}, q^{3/2}, \dots) s_{(\mu/\nu)^t}(q^{1/2}, q^{3/2}, \dots) , \tag{66}$$

where one should note that the representations are transposed due to the fact that $\Theta_3(e^{i\theta})$ is expressed in terms of $E(x; z)$ rather than $H(x; z)$.

Let us now consider what the assumptions of Szego's theorem imply for a function of the form $f(z) = E(x; z) E(x; z^{-1})$ or $f(z) = H(x; z) H(x; z^{-1})$. Let us consider first the case $f(z) = E(x; z) E(x; z^{-1})$. We repeat the top line of (123),

$$E(x; z) = \sum_{k=0}^\infty e_k(x) z^k = \prod_{k=1}^\infty (1 + x_k z) = \exp\left[ \sum_{k=1}^\infty (-1)^{k+1} \frac{p_k(x)}{k} z^k \right] , \tag{67}$$

so that

$$f(z) = \exp\left(\sum_{k=1}^{\infty} (-1)^{k+1} \frac{p_k(x)}{k}(z^k + z^{-k})\right) . \tag{68}$$

Therefore,

$$c_k = (-1)^{k+1} \frac{p_k(x)}{k} = c_{-k} , \quad k \neq 0 , \tag{69}$$

and (48) is written as

$$\sum_{k=1}^{\infty} \frac{|p_k(x)|}{k} < \infty , \quad \sum_{k=1}^{\infty} \frac{|p_k(x)|^2}{k} < \infty , \tag{70}$$

where we ignore an irrelevant factor 2. We see that

$$\lim_{k\to\infty} |p_k(x)| \to 0 , \tag{71}$$

as $\sum_{k=1}^{\infty} \frac{|p_k(x)|}{k}$ diverges otherwise. If we take $x_j$ to be real-valued, as we do in the explicit examples considered here, equation (71) requires that $x_j < 1$. The right requirement in (70) is strictly weaker than the left, so it does give rise to any additional restrictions. In the above expressions, if we replace $E(x;z)$ by $H(x;z)$, we have,

$$c_k = \frac{p_k(x)}{k} = c_{-k} , \quad k \neq 0 , \tag{72}$$

so that the assumptions of Szegö's theorem are given by (71) as well.

## 4.2 Finite $N$

Although the expressions given above were derived for $N \to \infty$, some of them can, in fact, be generalized to finite $N$ in case the number of distinct non-zero variables $x_j$ is smaller than $N$. From equations (60), (61), and (64), we see that, for finite $N$ and $f(z) = H(x;z)H(x;z^{-1})$,

$$\frac{\int s_\lambda(U)s_\mu(U^{-1})\tilde{f}(U)dU}{\int \tilde{f}(U)dU} = \frac{\sum_\kappa (s_\kappa(x))^2}{\sum_{\ell(\rho)\leqslant N}(s_\rho(x))^2} \sum_\nu s_{\lambda/\nu}(x)s_{\mu/\nu}(x) - \frac{\sum_{\ell(\nu)>N} s_{\nu/\lambda}(x)s_{\nu/\mu}(x)}{\sum_{\ell(\rho)\leqslant N}(s_\rho(x))^2} . \tag{73}$$

Let us denote the number of non-zero variables by $i_{\max}$, i.e. $x_i \neq 0$ for $i \leqslant i_{\max}$ and $x_i = 0$ for $i > i_{\max}$. In that case, $s_\kappa(x) = 0$ for $\ell(\kappa) > i_{\max}$, see equation (136), so that

$$\frac{\sum_\kappa (s_\kappa(x))^2}{\sum_{\ell(\rho)\leqslant N}(s_\rho(x))^2} = 1 . \tag{74}$$

Indeed, in case $N - \ell(\lambda) > i_{\max}$ and $N - \ell(\mu) > i_{\max}$, we can apply (136) again to find

$$\sum_{\ell(\nu)>N} s_{\nu/\lambda}(x)s_{\nu/\mu}(x) = 0 . \tag{75}$$

From this we conclude that, for $N - |\lambda| > i_{\max}$ and $N - |\mu| > i_{\max}$, we have

$$\frac{\int s_\lambda(U)s_\mu(U^{-1})\tilde{f}(U)dU}{\int \tilde{f}(U)dU} = \sum_\nu s_{\lambda/\nu}(x)s_{\mu/\nu}(x) , \tag{76}$$

i.e. the asymptotic expression (65) still holds in this case. Again, the above expressions still hold if we replace $H(x;z)$ by $E(x;z)$ and all representations by their transposes.

# 5 Spectral form factor

Although the main focus of this paper is the SFF of the $U(N)$ Chern-Simons matrix model, many of the techniques applied to this particular case can be applied to any function $f(z)$ satisfying the assumptions of Szegö's theorem [38], [39]. We will first keep the treatment general before considering the case $f(z) = \Theta_3(z)$.

## 5.1 The spectral form factor for general weight function

We repeat for convenience [62], [68]

$$\text{tr}U^n = \sum_\lambda \chi^\lambda_{(n)} s_\lambda(U) = \sum_{r=0}^{n-1} (-1)^r s_{(n-r,1^r)}(U) , \tag{77}$$

where we take $n \in \mathbb{Z}^+$. It is clear that this also holds when we replace $U$ with $U^{-1}$. Indeed, the expressions given below generalize to all integers if we replace $n$ by $|n|$ in the expressions below. The SFF is given by

$$NK(n) = \frac{1}{Z_N} \int dU \tilde{f}(U) \sum_{r,s=0}^{n-1} (-1)^{r+s} s_{(n-r,1^r)}(U^{-1}) s_{(n-s,1^s)}(U) , \tag{78}$$

where we remind the reader that $(n-r, 1^r)$ is a representation corresponding to a hook-shaped Young tableau with $n-r$ boxes in the first row and $r$ further rows with a single box. Writing $f(e^{i\theta}) = H(x; e^{i\theta})H(x; e^{-i\theta})$, we use (65) to find

$$NK(n) = \sum_\nu \sum_{r,s=0}^{n-1} (-1)^{r+s} s_{(n-r,1^r)/\nu}(x) s_{(n-s,1^s)/\nu}(x) , \quad n \in \mathbb{Z}\backslash\{0\} . \tag{79}$$

The first sum on the right hand side runs over all representations $\nu$ satisfying $\nu \subseteq (n-r, 1^r)$ as well as $\nu \subseteq (n-s, 1^s)$, so that $\nu = (a, 1^b)$ with $a \leqslant n-r, n-s$ and $b \leqslant r, s$. We remind the reader that (79) also holds when we replace $H(x; z)$ by $E(x; z)$ due to the fact that the SFF is invariant under transposition of the representations $(n-r, 1^r)$ and $(n-s, 1^s)$. There are three types of skew Schur polymomials $s_{\lambda/\mu}$ which appear in (79):

1. If $\nu = \lambda$, the skew Schur polynomial $s_{\lambda/\nu} = s_{\lambda/\lambda} = 1$.

2. If $\nu$ is the empty partition $\nu = \varnothing$, $s_{\lambda/\nu} = s_\lambda$ i.e. the skew Schur polynomials reduces to the usual (non-skew) Schur polynomial.

3. Then there is the case of two non-empty hook-shaped diagrams $\lambda = (n-r, 1^r)$ and $\nu = (a, 1^b)$ with $n-r > a$ and $r > b$ and $\nu$ non-empty, so that $\lambda/\nu$ consists of a row of $n-r-a$ boxes and a column of $r-b$ boxes. It is clear from equation (130) that the skew Schur polynomial factorizes as

$$s_{\lambda/\mu} = s_{(n-r-a)} s_{(1^{r-b})} = h_{n-r-a} e_{r-b} . \tag{80}$$

This can be made more clear using Young diagrams. Taking $n = 6$, $r = 2$ and $a = 2$, $b = 1$, equation (80) is given by the following, where one should keep in mind that the contributions corresponding to the two disconnected young diagrams are multiplied

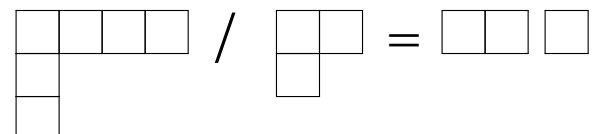

From the first point listed above, we see that there are $n$ terms in (79) with for which $\lambda = \mu = v = (n - r, 1^r)$. These terms give the following contribution

$$\sum_{r=0}^{n-1} \underbrace{s_\varnothing(U^{-1}) s_\varnothing(U)}_{=1} = n \ . \tag{81}$$

Perhaps surprisingly, we see from the above expression that terms satisfying $\lambda = \mu = v$ always reproduce the linear ramp of the CUE spectral form factor for $n \leqslant N$ (see e.g. (5.14.14) in [48]). It is well known that, for the CUE SFF, the linear ramp saturates at a plateau for $n \geqslant N$ [4], [48]. Here, too, the linear ramp gives way to a plateau, which comes about as follows. Remember that $s_\lambda(x)$ vanishes if the longest column in $\lambda$ contains more boxes than the number of non-zero variables in the set $x$ (136). We saw that we get a contribution equal to unity for every term for which $(n - r, 1^r) = v = (n - s, 1^s)$ for $0 \leqslant r \leqslant \text{Min}(n - 1, N - 1)$. However, there are only $N$ such reps, as $s_{(a,1^b)}(x) = 0$ if $b \geqslant N$. From this, we conclude that the contributions coming from $\lambda = v = \mu$ exactly reproduce the ramp and plateau.

Let us now consider those terms from which deviations from the linear ramp may arise. If $v$ is the empty set, as in point 2, we recover the disconnected part of the SFF, which is given by the square of

$$\sum_{r=0}^{n-1} (-1)^r s_{(n-r,1^r)}(x) = \langle \text{tr} U^n \rangle \ . \tag{82}$$

The remaining terms, coming from point 3, is given by the square of

$$\sum_{r=0}^{n-1} \sum_{\substack{v \neq \varnothing \\ v \neq (n-r,1^r)}} (-1)^r s_{(n-r,1^r)/v}(x) \ . \tag{83}$$

At first sight, this may seem like a rather rather complicated expression. Let us factor the expression in (83) into two separate sums over $r$ and $s$ and consider one such sum for a fixed choice of $v = (1)$. Remembering that $s_{(n-r,1^r)/(1)} = h_{n-r-1} e_r$ and using equation (2.6') of [58], we find for a single such sum,

$$\sum_{r=0}^{n-1} (-1)^r s_{(n-r,1^r)/(1)} = \sum_{r=0}^{n-1} (-1)^r h_{n-r-1} e_r = 0 \ . \tag{84}$$

Taking $n = 4$, the above identity can be expressed in terms of Young diagrams as follows.



The identity $\sum_{r=0}^{n-1} (-1)^r h_{n-r-1} e_r = 0$ can be seen from $H(x; t) E(x; -t) = 1$, see equation (123). Equation (84) can then be found by checking every order of $t$ in $H(x; t) E(x; -t)$. One can see from these considerations that any term corresponding to a single choice of $v$ in (83) is equal to zero. The contribution for general $v = (a, 1^b) \subset (n - r, 1^r)$ with $v \neq \varnothing$ and $v \neq (n - r, 1^r)$ is given by

$$\sum_{r=b}^{n-a} (-1)^r h_{n-r-a} e_{r-b} = \sum_{r=0}^{n-b-a} (-1)^r h_{n-b-a-r} e_r = 0 \ . \tag{85}$$

In short, the contribution arising from $\nu \neq \varnothing, (n-r, 1^r)$ is equal to zero.

We now compute the explicit expression for the disconnected SFF. Applying (66) with $\mu = \varnothing$, we have,

$$\langle \text{tr} U^n \rangle = p_n(x) = \sum_i x_i^n \ . \tag{86}$$

The functions $p_n(x)$ are the power-sum polynomials, mentioned in appendix B. The fact that we get power-sum polynomials should not be surprising due to the statements below equations (44) and (65). Namely, $\text{tr} U^n$ is the $n^{\text{th}}$ power sum polynomial in the eigenvalues of $U$, and the evaluation of the matrix integral of a single matrix trace amounts to replacing the eigenvalues by the variables $x_i$, which immediately leads to (86). Below equation (70), we show that the assumptions of Szegő's theorem require

$$\lim_{k \to \infty} p_k(x) = 0 \ , \tag{87}$$

so that the disconnected part of the SFF goes to zero. Hence, we see that the plateau of the SFF is exact, that is

$$\lim_{n \to \infty} K(n) = 1 \ . \tag{88}$$

We thus find,

$$K(n) = \begin{cases} \frac{1}{N} \left[ n + p_n(x)^2 \right] \ , & n/N \leqslant 1 \ , \\ 1 & , \ n/N \geqslant 1 \ . \end{cases} \tag{89}$$

This is the main result of the present work.

Let us now give some basic expressions for the SFF itself. From the form of (89), we can give an expression for the behaviour of the SFF upon rescaling $x_i$. The linear ramp remains unaffected by rescaling as it is independent of choice of variables $x_i$. Further, since $\langle \text{tr} U^n \rangle = \sum_{r=0}^{n-1} (-1)^r s_{(n-r, 1^r)}(x)$ is a sum of polynomials of degree $n$ in $x_i$, we have upon rescaling as $x_j \mapsto A x_j$, where $A$ is some number,

$$p_n(Ax) = A^n p_n(x) \ . \tag{90}$$

Further, we take $x_j \mapsto (x_j)^k$ with $k \in \mathbb{Z}^+$, we have, writing $x^k = (x_1^k, x_2^k, \dots)$,

$$p_n(x^k) = p_{kn}(x) \ . \tag{91}$$

This naturally generalizes to $k \in \mathbb{R}$ if we take the label $n$ of $p_n(x)$ to be a general real number. We plot an example of an SFF in figure 1. In the figure, we indicate lines with $kn =$ constant, which lie at 45 degrees. Although this SFF was computed for a specific choice of weight function, it follows from equation (91) that lines of constant $kn$ always lie at 45 degrees. The linear ramp then corresponds to $kn \to \infty$.

For a finite number of variables, the calculation of the SFF from (89) is rather straightforward. In case we have a very large number of non-zero variables, $p_n(x)$ is generally rather hard to calculate, except for certain known examples. Let us take $x_k = 1/(k+1)^2$. Using the well-known product expansion of the hyperbolic sine as $\sinh(\pi t) = \pi t \prod_{k \geqslant 1} \left(1 + \frac{t^2}{k^2}\right)$, we have

$$f(z) = \prod_{k=1}^{\infty} \left(1 + \frac{z}{(k+1)^2}\right) \left(1 + \frac{z^{-1}}{(k+1)^2}\right) = \frac{\sinh(\pi z^{1/2}) \sinh(\pi z^{-1/2})}{\pi^2 (2 + z + z^{-1})} \ . \tag{92}$$

Further, we have,

$$p_n(x) = \sum_{k=1}^{\infty} \frac{1}{(k+1)^{2n}} = \zeta(2n) - 1 \ , \tag{93}$$

where $\zeta(s)$ is the Riemann zeta function. The SFF for weight function (92) is therefore given by

$$NK(n) = \begin{cases} n + (\zeta(2n)-1)^2 & , \quad n \leqslant N \,, \\ N & , \quad n \geqslant N \,. \end{cases} \tag{94}$$

### 5.1.1 General trace identities

We now consider some expectation values of $\mathrm{tr}U^n$ with some more general objects. For example we can conclude from the arguments leading to (89) that the connected part of $\langle \mathrm{tr}U^n \mathrm{tr}U^{-k} \rangle$, for $k, n \in \mathbb{Z}^+$, is given by

$$\langle \mathrm{tr}U^n \mathrm{tr}U^{-k} \rangle_c = \sum_{s=0}^{k-1} \sum_{r=0}^{n-1} \sum_{v \neq \varnothing} (-1)^{r+s} s_{(k-s,1^s)/v} s_{(n-r,1^r)/v} = n\delta_{nk} \,. \tag{95}$$

In particular, let us take $k < n$. In that case, any $v \in (k-s, 1^s)$ for all $s \in \{0, \ldots, k-1\}$ necessarily satisfies $|v| \leqslant k < n$, so that $(n-r, 1^r)/v \neq \varnothing$ for any partition $(n-r, 1^r)$, $r \in \{0, \ldots, n-1\}$. Using (85), the result is again zero. Note that equation (95) can easily be found for the CUE case by using bosonization [69][70]. More generally, let us consider expectation values of the form

$$\langle \mathrm{tr}U^{-n} \mathrm{tr}_\lambda U \rangle_c = \sum_{v \neq \varnothing} \sum_{r=0}^{n-1} (-1)^r s_{(n-r,1^r)/v} \, s_{\lambda/v} \,. \tag{96}$$

Since fixing any $v \subseteq (n-r, 1^r)$ in (96) with $v \neq (n-r, 1^r)$ gives zero upon summing over $r$, we only get a nonzero answer for terms for which $v = (n-1, 1^r) \subseteq \lambda$. That is,

$$\langle \mathrm{tr}U^{-n} \mathrm{tr}_\lambda U \rangle_c = \sum_{r=\min(0,n-\lambda_1)}^{\min(n-1,\lambda_1^t+1)} (-1)^r s_{\lambda/(n-r,1^r)} \,, \tag{97}$$

where the boundaries on the sum arise from the fact that we only sum over those representations $(n-r, 1^r)$ which satisfy $(n-r, 1^r) \subseteq \lambda$. Equation (97) greatly simplifies certain calculations. For example, consider $(n-r, 1^r) \not\subseteq \lambda \; \forall \; r \in \{0, \ldots, n-1\}$. Another way to write this is that $\lambda_1 + \lambda_1^t - 1 < n$. We then have,

$$\langle \mathrm{tr}U^{-n} \mathrm{tr}_\lambda U \rangle_c = 0 \,. \tag{98}$$

Let us represent $\lambda$ in Frobenius notation as $\lambda = (a_1, \ldots, a_k | b_1, \ldots, b_k)$ with $a_i$ and $b_j$ non-negative integers satisfying $a_1 > \cdots > a_k$ and $b_1 > \cdots > b_k$. In this case, $a_1 + b_1 + 1$ gives the number of boxes in the upper left hook of $\lambda$, or, equivalently, the hook-length of the top left box in $\lambda$, labelled by $x = (1,1)$ in the notation of appendix C. In this notation, (98) states that $\langle \mathrm{tr}U^{-n} \mathrm{tr}_\lambda U \rangle_c = 0$ if $a_1 + b_1 + 1 < n$. For the specific case of the Chern-Simons matrix model this identity has an interesting interpretation which we comment on in section 5.2.

We can find similar identities for certain representations $\lambda$ with $a_1 + b_1 + 1 > n$. Define $m := a_1 + b_1 + 1 - n$ and consider $\lambda = (a|b) = (a_1, \ldots, a_k | b_1, \ldots b_k)$ satisfying $m \leqslant a_1 - a_2 - 1$ and $m \leqslant b_1 - b_2 - 1$, or, equivalently, $m \leqslant \lambda_1 - \lambda_2$ and $m \leqslant \lambda_1^t - \lambda_2^t$, respectively. Let us take $\mu = (a_2, \ldots, a_k | b_2, \ldots, b_k)$, which is constructed from $\lambda$ by removing the first row and column. For any rep $(n-r, 1^r)$ satisfying $(n-r, 1^r) \subseteq \lambda$, we then have

$$\lambda/(n-r,1^r) = (a_1+1,1^{b_1})/(n-r,1^r) \times \mu = (a_1+1-n+r) \times (1^{b_1-r}) \times \mu \,. \tag{99}$$

That is, $\lambda/(n-r, 1^r)$ factorizes as the skew partition of two hook shapes times the partition obtained from $\lambda$ by deleting the top-left hook. In terms of Young diagrams, an example is given by the following.

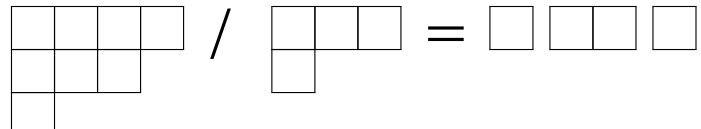

Since $(a_1 + 1, 1^{b_1})/(n - r, 1^r)$ is a product of a row and a column, we can again use (84) to find

$$
\begin{aligned}
\left\langle \mathrm{tr} U^{-n} \mathrm{tr}_\lambda U \right\rangle_c &= \sum_{r=n-a_1-1}^{b_1} (-1)^r s_{\lambda/(n-r,1^r)} \\
&= (-1)^{n-a_1-1} s_\mu \sum_{k=0}^{m} (-1)^k h_{m-k} e_k = 0 \, .
\end{aligned}
\tag{100}
$$

## 5.2 The SFF of the Chern-Simons matrix model

As noted before, the SFF of the Chern-Simons matrix model corresponds to a $(2n, 2)$-torus link with one component in the fundamental and the other in the antifundamental representation. Whereas expressions for link invariants of the form $\left\langle \mathrm{tr} U^{n_1} \mathrm{tr} U^{n_2} \dots \right\rangle$ with $n_i \geqslant 2$ have appeared in the literature [68], [71], [72], expressions with powers of mixed signature, to the best of the authors' knowledge, have not. The expressions presented in the previous section allow us to calculate precisely those objects. In particular, the SFF, is again given by (89). We can easily calculate the non-trivial part of the SFF, $\left\langle \mathrm{tr} U^n \right\rangle^2$, for $|q| < 1$, by using the expression in terms of power-sum polynomials. However, it is instructive to see how this arises from the functional form of this object as a function of $N$, before taking $N \to \infty$. This is particularly useful in knot theory, as the expression for general $N$ may allows one to distinguish various knots and links which may have the same invariant when one ignores the dependence on $N$. Let us apply (138) to the hook-shaped representation $(a, 1^b)$, which gives the following expression for general $N$

$$
s_{(a,1^b)}(x_i = q^{i-1}) = q^{\frac{1}{2} b(b+1)} \frac{[N + a - 1]!}{[N - b - 1]![a - 1]![b]![a + b]} \, .
\tag{101}
$$

We now use (77) and (66) to calculate $\left\langle \mathrm{tr} U^n \right\rangle$, which, for the lowest values of $n$, is given by

$$
\begin{aligned}
\left\langle \mathrm{tr} U \right\rangle &= q^{1/2} \frac{1 - q^N}{1 - q} = q^{1/2} [N]_q \, , \\
\left\langle \mathrm{tr} U^2 \right\rangle &= \frac{q(1 - q^{2N})}{1 - q^2} = q [N]_{q^2} \, , \\
\left\langle \mathrm{tr} U^3 \right\rangle &= \frac{q^{3/2}(1 - q^{3N})}{1 - q^3} = q^{3/2} [N]_{q^3} \, .
\end{aligned}
\tag{102}
$$

One can see a simple pattern emerge in (102). Indeed, using (77) and taking into account the comments made below (65), we see that

$$
\left\langle \mathrm{tr} U^n \right\rangle = p_n(x_j = q^{j-1/2}) = q^{n/2} \sum_{j=1}^{N} q^{n(j-1)} = q^{n/2} \frac{1 - q^{nN}}{1 - q^n} = q^{n/2} [N]_{q^n} \, .
\tag{103}
$$

That is, the asymptotic $(n, 1)$-torus knot invariant is given by the $q^n$-deformation of $N$ times a factor $q^{n/2}$. As far as the authors are aware, this statement has heretofore not appeared in the literature.

As mentioned above, as well as in appendix C, the limit $N \to \infty$ simplifies these expressions even further. Upon this simplification, the final expression for the SFF is then given by

$$NK(n) = \begin{cases} n + (q^n + q^{-n} - 2)^{-1} \ , & n \leqslant N \ , \\ N & , \quad n \geqslant N \ . \end{cases} \tag{104}$$

The SFF is plotted in 1 for $n = 1, \ldots, 20$, with $q = 0, 9^k$, $k = 1, \ldots, 9$.

### 5.2.1 General identities for the Chern-Simons matrix model

The identities we derived in 5.1.1 apply to the Chern-Simons matrix model as well, in which case they have an interpretation in terms of knot and link invariants. For example, take (98), which says that, for $\lambda$ satisfying $(n - 1, 1^r) \nsubseteq (n - r, 1^r) \ \forall r \in \{1, \ldots, n - 1\}$,

$$\left\langle \mathrm{tr} U^{-n} \mathrm{tr}_\lambda U \right\rangle_c = 0 \quad \Rightarrow \quad \left\langle \mathrm{tr} U^{-n} \mathrm{tr}_\lambda U \right\rangle = \left\langle \mathrm{tr} U^{-n} \right\rangle \left\langle \mathrm{tr}_\lambda U \right\rangle \ . \tag{105}$$

In terms of knot and link invariants, the above expression entails that expectation value of the product of an $(n, 1)$-torus link with an unknot in representation $\lambda$ with opposite orientation equals the product of their expectation values.

Another trace identity derived in 5.1.1 is equation (105). This equation expresses the fact that a Wilson line in the (anti)fundamental rep winding $n$ times around a article in rep some $\lambda$ will give a vanishing connected expectation value if the $\lambda_1 - \lambda_1^t - 1 - n \leqslant \lambda_1 - \lambda_2$ and $\lambda_1 - \lambda_1^t - 1 - n \leqslant \lambda_1^t - \lambda_2^t$. Further, it is worth emphasizing that, using (97), one can calculate pretty much any object of the form

$$\left\langle \mathrm{tr} U^{-n} \mathrm{tr}_\lambda U \right\rangle \ , \tag{106}$$

as all the objects appearing on the right hand side of the above expression are skew Schur polynomials with variables $x_i = q^{i-1/2}$, to which we can apply the $q$-hook length formula in equation (138).

## 6 Overview and Conclusions

Here, we put forward a conjecture that many, if not all, examples of invariant one-matrix models which exhibit intermediate statistics are given by matrix models of topological field or string theories. We explicitly support this conjecture by the example of the matrix model introduced in [13], which is the matrix model of $U(N)$ Chern-Simons model on $S^3$. The latter model is directly related to A and B topological string models via the Gopakumar-Vafa duality.

To calculate the SFF of the Chern-Simons matrix model, we consider general infinite order unitary matrix models with weight functions satisfying the assumptions of Szegö's theorem. We find that the SFF's for these models have a surprisingly concise form, with the connected SFF giving rise to the linear ramp and plateau, while the disconnected part gives rise to a dip. Moreover, from the assumptions of Szegö's theorem, it follows that the dip had to go to zero, so that the plateau is exact. Further, we derive certain identities on expectation values of products of traces, as well as the behavior of the SFF under certain changes of the weight function.

We then apply these general results to the matrix model for $U(N)$ Chern-Simons theory for $S^3$, studied by Muttalib and collaborators for its intermediate statistics. The SFF of this model is a topological (link) invariant. In particular, it is given by the HOMFLY invariant $(2n, 2)$-torus links with one component in the fundamental and the other in the antifundamental

representation, an explicit expression of which, to the best of the authors knowledge, did not appear in the literature before. It displays the hallmark characteristics of intermediate statistics, with a dip that becomes more pronounced as we move further away from the CUE limit, $q \to 0$. One can identify various matrix models which have the same SFF, an immediate example of which is given by replacing $E(x;z)$ by $H(x;z)$.

The present work provides the tools to shed more light on the connections between topological field theories and intermediate statistics; we believe that the matrix models which arise in topological string theory are natural tools for describing ergodic-to-nonergodic phase transitions. Indeed, this paper provides a first example of what we suspect to be a broader connection between intermediate statistics and topology.

# Acknowledgements

We would like to thank Wouter Buijsman, Oleksandr Gamayun, Alex Garkun, and Miguel Tierz for valuable discussions and comments, and Vladimir Kravtsov for inspiring this study and useful discussions.

**Funding information** This work is part of the DeltaITP consortium, a program of the Netherlands Organization for Scientific Research (NWO) funded by the Dutch Ministry of Education, Culture and Science (OCW).

# A $q$-Numbers

We review some basic facts and useful relations involving $q$-numbers, which are so-called *q-deformations* of more familiar (generally complex) numbers. We will only be considering $q$-deformation of positive integers here, which are defined as

$$[n]_q = (1 + q + \cdots + q^{n-1}) = \frac{1 - q^n}{1 - q} \ , \quad n \in \mathbb{Z}^+ \ . \tag{107}$$

Other definitions of $[n]_q$, such as $\frac{q^{-n/2} - q^{n/2}}{q^{-1/2} - q^{1/2}}$, also appear in the literature. Their common feature is that

$$\lim_{q \to 1^-} [n]_q = n \ . \tag{108}$$

Note that, for $k, m, n \in \mathbb{Z}^+$ satisfying $\frac{m}{n} = k$, we have

$$\frac{[m]_q}{[n]_q} = [k]_{q^n} \ , \quad \frac{[r \cdot m]}{[r \cdot n]} = [k]_{q^{nr}} \ , \tag{109}$$

for example,

$$\frac{[8]_q}{[2]_q} = \frac{1 + q + \cdots + q^7}{1 + q} = 1 + q^2 + q^4 + q^6 = [4]_{q^2} \ . \tag{110}$$

We will write $[n]_q$ as $[n]$ henceforth and only specify the deformation parameter in case it is different from $q$. $q$-Factorials and $q$-binomials are defined as follows. For $n, k \in \mathbb{Z}^+$

$$[N]! = (1 + q)(1 + q + q^2) \ldots (1 + q + \cdots + q^{N-1}) \ , \quad \begin{bmatrix} N \\ k \end{bmatrix} = \frac{[N]!}{[N-k]![k]!} \ . \tag{111}$$

We then introduce the *q-Pochhammer symbol,* which is defined as

$$(a;q)_k = (1-a)(1-aq)\ldots(1-aq^{k-1}) \,. \tag{112}$$

Note that

$$(a;q)_n = \frac{(a;q)_\infty}{(aq^n;q)_\infty} \,. \tag{113}$$

Note also that

$$[n]! = \frac{(q;q)_n}{(1-q)^n} \,, \tag{114}$$

from which follows

$$\begin{bmatrix} N \\ k \end{bmatrix} = \frac{(q;q)_N}{(q;q)_{N-k}(q;q)_k} = \frac{(1-q^N)(1-q^{N-1})\ldots(1-q^{N-r+1})}{(1-q)(1-q^2)\ldots(1-q^k)} \,. \tag{115}$$

We see from this expression that, for $q < 1$, we have

$$\lim_{N\to\infty} \begin{bmatrix} N \\ k \end{bmatrix} = \frac{1}{(q;q)_k} \,, \tag{116}$$

$q$-Pochhammer symbols can be generalized as follows

$$(a_1, a_2, \ldots, a_m; q)_n = \prod_{j=1}^{m} (a_j; q)_n \,. \tag{117}$$

These are rather versatile objects. For example, Jacobi's third theta function can be expressed through the Jacobi triple product as

$$\sum_{n\in\mathbb{Z}} q^{n^2/2} z^n = (q, -q^{1/2}z, -q^{1/2}/z; q)_\infty \,, \quad 0 < |q| < 1 \,. \tag{118}$$

Note that that the definition in (118) has $q^{n^2/2}$ rather than $q^{n^2}$ as expansion coefficients, following the convention of e.g. [34]. This is the origin of the differences with the expressions appearing e.g. in [68], which are related to the expressions given here by taking $q \to q^2$.

## B  Symmetric polynomials

We review here some basic aspects of symmetric polynomials in the set of variables $x = (x_1, x_2, \ldots)$. The *elementary symmetric polynomials* are then defined as

$$e_k(x) = \sum_{i_1 < \cdots < i_k} x_{i_1} \ldots x_{i_k} \,. \tag{119}$$

Some examples include

$$e_0 = 1 \,,$$
$$e_1(x_1) = x_1 \,,$$
$$e_1(x_1, x_2) = x_1 + x_2 \,,$$
$$e_2(x_1, x_2) = x_1 x_2 \,.$$

Closely related are the *complete homogeneous symmetric polynomials,* defined as

$$h_k(x) = \sum_{i_1 \leqslant \cdots \leqslant i_k} x_{i_1} \ldots x_{i_k} \,, \tag{120}$$

which contains all monomials of degree $j$. Note the difference in the summation bounds between (119) and (120). Some examples of these include

$$h_0 = 1 \, ,$$
$$h_1(x_1) = x_1 \, ,$$
$$h_1(x_1, x_2) = x_1 + x_2 \, ,$$
$$h_2(x_1, x_2) = x_1^2 + x_2^2 + x_1 x_2 \, .$$

Another example is the set of *power-sum symmetric polynomials*,

$$p_k(x) = x_1^k + x_2^k + \ldots \, . \tag{121}$$

Note that if a matrix $U$ has $d_i$ as its eigenvalues, traces of moments of $U$ are given by power-sum symmetric polynomials, that is,

$$\mathrm{tr}U^k = p_k(d) \, . \tag{122}$$

Defining $z = e^{i\theta}$ as in (46), we have the following relations between the above polynomials [58]

$$E(x; z) = \sum_{k=0}^{\infty} e_k(x) z^k = \prod_{k=1}^{\infty} (1 + x_k z) = \exp\left[ \sum_{k=1}^{\infty} \frac{(-1)^{k+1}}{k} p_k(x) z^k \right] \, ,$$

$$H(x; z) = \sum_{k=0}^{\infty} h_k(x) z^k = \prod_{k=1}^{\infty} \frac{1}{1 - x_k z} = \exp\left[ \sum_{k=1}^{\infty} \frac{1}{k} p_k(x) z^k \right] \, . \tag{123}$$

Consider the example where $x_i = q^{i-1}$, so that (see [58] I.2 examples 3 and 4)

$$E(t) = \prod_{i=0}^{N-1} (1 + q^i t) = \sum_{k=0}^{N} q^{k(k-1)/2} \begin{bmatrix} N \\ k \end{bmatrix} t^k \, . \tag{124}$$

Similarly,

$$H(t) = \prod_{i=0}^{N-1} (1 - q^i t)^{-1} = \sum_{k=0}^{N} \begin{bmatrix} N + k - 1 \\ k \end{bmatrix} t^k \, , \tag{125}$$

so that

$$e_k = q^{k(k-1)/2} \begin{bmatrix} N \\ k \end{bmatrix} \, , \quad h_k = \begin{bmatrix} N + k - 1 \\ k \end{bmatrix} \, . \tag{126}$$

Here, $e_k$ is only defined for $k \leqslant N$. From (116), we see that, for $q < 1$ and $N \to \infty$,

$$e_k = \frac{q^{k(k-1)/2}}{(q; q)_k} \, , \quad h_k = \frac{1}{(q; q)_k} \, . \tag{127}$$

## C  Schur polynomials

A somewhat less straightforward type of symmetric polynomial is the *Schur polynomial*, which reduces to some of the above examples in certain cases. Schur polynomials play an important role as characters of irreducible representations, often referred to as irreps, of general linear groups and subgroups thereof. Irreps can be conveniently classified by partitions, and we use these terms interchangably in this work. We denote partitions as $\lambda = (\lambda_1, \lambda_2, \ldots, \lambda_\ell)$, which are sequences of non-negative integers ordered as $\lambda_1 \geqslant \lambda_2 \geqslant \ldots$ Typically, partitions are

taken to have a finite number of elements, that is, only a finite number of $\lambda_i$ are non-zero, but we will impose no such restriction. The *weight* of a partition (not to be confused with the highest weight of the corresponding irrep) is given by the sum of its terms $|\lambda| = \sum_i \lambda_i$ and its *length* $\ell(\lambda)$ is the largest value of $i$ for which $\lambda_i \neq 0$. A *semistandard Young tableau* (SSYT) corresponding to $\lambda$ is then given by positive integers $T_{i,j}$ satisfying $1 \leqslant i \leqslant \ell(\lambda)$ and $1 \leqslant j \leqslant \lambda_i$. These integers are required to increase weakly along every row and increase strongly along every column, i.e. $T_{i,j} \geqslant T_{i,j+1}$ and $T_{i,j} > T_{i+1,j}$ for all $i, j$. Label by $\alpha_i$ the number of times that the number $i$ appears in the SSYT. We then define

$$x^T = x_1^{\alpha_1} x_2^{\alpha_2} \dots . \tag{128}$$

The *Schur polynomial* $s_\lambda(x)$ is given by [59].

$$s_\lambda(x) = \sum_T x^T , \tag{129}$$

where the sum runs over all SSYT's corresponding to $\lambda$ i.e. all possible ways to inscribe the diagram corresponding to $\lambda$ with positive integers that increase weakly along rows and strictly along columns. We give an example of an SSYT corresponding to a Young diagram $\lambda = (3, 2)$. From (131) one can see that the contribution of the SSYT below would be given by $x_1^2 x_2 x_3^2$.

| 1 | 1 | 3 |
|---|---|---|
| 2 | 3 |   |

We can see from the above definition that

$$s_{(1^n)} = e_n \ , \quad s_{(n)} = h_n , \tag{130}$$

i.e. the Schur polynomial of a column or row of $n$ boxes is given by a degree $n$ elementary or homogeneous symmetric polynomial, respectively. Schur polynomials have a natural generalization to so-called *skew Schur polynomials*. In this case we have two diagrams $\lambda$ and $\mu$ such that $\mu \subseteq \lambda$ i.e. $\mu_i \leqslant \lambda_i, \ \forall \ i$. We denote by $\lambda/\mu$ the complement of $\mu$ in the diagram corresponding to $\lambda$. Define a semistandard skew Young tableau corresponding to $\lambda/\mu$ similar to the above, namely, as an array of positive integers $T_{ij}$ satisfying $1 \leqslant i \leqslant \ell(\lambda)$ and $\mu_i \leqslant j \leqslant \lambda_i$ which increase weakly along rows and strictly along columns. We then define the *skew Schur polynomial* corresponding to $\lambda/\mu$ as

$$s_{\lambda/\mu} = \sum_T x^T, \tag{131}$$

where the sum again runs over all SSYT's corresponding to $\lambda/\mu$. Note that if $\mu$ is the empty partition, i.e. $\mu_i = 0, \ \forall i$, we have $s_{\lambda/\mu} = s_\lambda$, and if $\lambda = \mu$, $s_{\mu/\mu} = 1$. Let us consider $\lambda = (3, 2)$ and $\mu = (1)$. Below, we give an SSYT corresponding to the skew partition $\lambda/\mu$, which would contribute $x_1^2 x_2 x_3$ to the skew Schur polynomial.

|   | 1 | 3 |
|---|---|---|
| 1 | 2 |   |

Skew Schur polynomials can also be expressed in determinantal form. Using a matrix of the form $\mathcal{M} = (x_j^{(N-k)})_{j,k=1}^N$, we have the following expression for the Vandermonde determinant

$$\det(x_j^{(N-k)})_{j,k=1}^N = \prod_{1 \leqslant j < k \leqslant N} (x_j - x_k) . \tag{132}$$

We then have

$$s_\lambda(U) = s_\lambda(x_j) = \frac{\det\left(x_j^{N-k+\lambda_k}\right)_{j,k=1}^N}{\det\left(x_j^{N-k}\right)_{j,k=1}^N}. \tag{133}$$

The (skew) Schur polynomials can be expressed in terms of elementary symmetric polynomials $e_k(x)$ or complete homogeneous symmetric polynomials $h_k(x)$ via the following determinantal expressions, known as the Jacobi-Trudi identities,

$$s_{(\mu/\lambda)} = \det(h_{\mu_j-\lambda_k-j+k})_{j,k=1}^{\ell(\lambda)} = \det(e_{\mu_j^t-\lambda_k^t-j+k})_{j,k=1}^{\lambda_1} = D_N^{\lambda,\mu}(H(x;z)),$$
$$s_{(\mu/\lambda)^t} = \det(e_{\mu_j-\lambda_k-j+k})_{j,k=1}^{\ell(\lambda)} = \det(h_{\mu_j^t-\lambda_k^t-j+k})_{j,k=1}^{\lambda_1} = D_N^{\lambda,\mu}(E(x;z)), \tag{134}$$

where the partition $\lambda^t$ is obtained from $\lambda$ by transposing the corresponding Young tableau and where $\varnothing$ refers to the empty partition. The objects on the right hand side of (134) are explained in section 4. Schur polynomials satisfy various useful identities, including the so-called Cauchy identity and its dual,

$$\sum_\lambda s_\lambda(x)s_\lambda(y) = \prod_{i=1,j=1}^\infty \frac{1}{1-x_iy_j}, \quad \sum_\lambda s_\lambda(x)s_{\lambda^t}(y) = \prod_{i=1,j=1}^\infty 1-x_iy_j. \tag{135}$$

Other useful identities for our purposes are the following, which can be found in Chapter I.5 of [58],

$$s_{\lambda/\mu}(x_1,\ldots,x_n) = 0 \text{ unless } 0 \leqslant \lambda_i^t - \mu_i^t \leqslant n \text{ for all } i \geqslant 1. \tag{136}$$

Note that an example of (136) is given by the fact that $e_k(x_1,\ldots,x_N) = 0$ for $k > N$. We consider some Schur polynomials which are treated in I.3 examples 1-4 of [58]. Schur polynomials with all variables equal to 1 give the hook-length formula for the dimension of the representation, that is

$$s_\lambda(1,\ldots,1) = \prod_{x\in\lambda} \frac{N+c(x)}{h(x)} =: \dim(\lambda), \tag{137}$$

where $c(x) = j - i$ for $x = (i,j) \in \lambda$ is the content of $x \in \lambda$, $h(i,j) = \lambda_i + \lambda_j^t - i - j + 1$ is its hook-length, and $n(\lambda) = \sum_i(i-1)\lambda_i$. If, instead, we choose variables as $x_i = q^{i-1}$, we get the following $q$-deformation of the dimension of $\lambda$

$$s_\lambda(x_i = q^{i-1}) = q^{n(\lambda)} \prod_{x\in\lambda} \frac{[N+c(x)]}{[h(x)]} =: q^{n(\lambda)} \dim_q(\lambda). \tag{138}$$

The quantity $\dim_q(\lambda)$ is known as the *quantum dimension*, or $q$-dimension. It is given by the hook length formula (137) where numbers are replaced by $q$-numbers. If we consider knots and links as consisting of the world lines of anyons carrying some representations $\lambda,\ldots$, $\dim_q(\lambda)$ gives the dimension of the Hilbert space of $\lambda$ [73]. Note that (109) implies that irreps with the same dimension can have different quantum dimensions. This is why Chern-Simons theory with $0 < q < 1$ can distinguish between certain (un)knots which give identical results in the limit $q \to 1$ or $q \to 0$.

In fact, the above expression simplifies even further. In particular, one can easily see that, for $|q| < 1$ and $N \to \infty$, quantum dimensions for reps with finite column lengths depend only on the hook lengths. This is because $q^{N-k} = 0$ for $k$ finite, so that, for $\lambda$ such that $c(x)$ is finite for all $x \in \lambda$,

$$\prod_{x\in\lambda} \frac{[N+c(x)]}{[h(x)]} = \frac{1}{(1-q)^{|\lambda|}} \prod_{x\in\lambda} \frac{1}{[h(x)]}. \tag{139}$$

In fact, since the Jacobi triple product expansion is only valid in case $0 < |q| < 1$ and we take $N \to \infty$ here, we see that the numerical values of the Schur polynomials considered here only depend on the hook-lengths of their components. Of course, one can still use the full functional form involving terms of the form $q^N$ to in the context of knot theory, as these functional forms can allow one to distinguish between knots or links which have the same hook-lengths. For example, the quantum dimensions of $(2)$ and $(1^2)$ are different when taking into account their dependence on $N$. On the other hand, these quantum dimensions are identical when we take into account the fact that $q^N = 0$ for $|q| < 1$ and $N \to \infty$. Lastly, one should note that, since the hook-lengths are invariant under transposition, the quantum dimensions involved are invariant under transposition as well.

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
