# Peer review of "Topological field theory approach to intermediate statistics"

_SciPost Physics, doi:SciPost Phys. 10, 146 (2021)_

## Round 1 · Referee Report · Anonymous (Referee 1) · 2021-2-23

Report
NOTE: The following is from an invited referee who submitted the report by email rather than through the web interface.
At the point of the Anderson transition a system exhibits an "intermediate statistics", i.e. intermediate between Wigner-Dyson (ergodic phase) and Poisson (nonergodic phase). This statistics is universal, although there are several universality classes. The authors propose a correspondence between these universality classes and some "interacting topological states of matter" which occur in the Chern-Simons or string theories. As far as I can see, their logic goes like this: (i) Anderson transition is described by certain matrix ensembles. (ii) These ensembles resemble matrices which appear in some topological phases in string theory. Ergo, (iii) These very complicated phases are in correspondence with the relatively simple "intermediate statistics" of the Anderson transition (the latter is a single particle problem and in this sense it is "simple").
I have a problem with statement (i) in this logical chain. It is true that a number of authors have proposed various matrix ensembles which, supposedly, describe the Anderson transition. Those proposals, however, have little to do with the genuine Anderson transition and its intermediate statistics and, at best, can be considered as some "toy models". Indeed, the spatial dimensionality, d, plays a decisive role in the Anderson transition (the transition exists only for d>2 and the statistics is sensitive to d). But there is no "d" either in the banded matrices or in the log-Gaussian ensemble mentioned by the authors. Moreover, those ensembles don't even exhibit an Anderson transition in its usual sense, i.e. as a transition in the energy band when the energy crosses some critical value. What happens in those ensembles is that, when some parameter is changed, the eigenfunctions of the entire spectrum are changing from being localized to being extended. A simple way to see that The Anderson transition has little to do with those ensembles is simply to write down the random matrix for the original Anderson model, call it the Anderson ensemble (AE). The AE is NOT invariant under an orthogonal transformations, neither it has any similarity with the banded matrix ensemble. The main property of the AE is sparsity (the matrices in 3d are more sparse than those in 2d). Thus, I would argue that sparsity is the main property which determines the existence of the transition and the corresponding intermediate statistics.
In short, the whole discussion of the Anderson transition and the intermediate statistics in the paper is superficial and sometimes simply wrong. For instance, the statement "An important example of such a system is given by disordered conductors, where increasing the disorder strength leads to greater deviation from Wigner-Dyson universality" is incorrect. There is no gradual change of the Wigner-Dyson statistics with the increase of disorder. It holds all the way down to the mobility edge E_c, while below E_c the Poisson statistics sets in. The intermediate statistics takes place only at E_c (of course, one should carefully define the limiting procedure, starting with the large finite sample, where E_c is smeared into a narrow interval which however contains a macroscopic number of levels).
All of the above doesn't imply that the paper is useless. Let's forget the Anderson transition. Then, the main content of the paper is the above statement (ii), namely, that there exists a correspondence between matrix ensembles coming from different fields, and that perhaps topological phases in string theory can be understood with the help of some simple, previously studied, matrix ensembles. This might well be true, although it's not for me to judge since I don't know anything about those topological phases in string theory.
Requested changes
At the very least, in their discussion of the Anderson localization and the matrix ensembles the authors should clearly state that those ensembles do not really describe the genuine Anderson problem but can merely serve as toy models for that phenomenon.
Author: Ward Vleeshouwers on 2021-03-03 [id 1282]
(in reply to Report 1 on 2021-02-23)
Below is our answer to the referee report from 23-02-'21. We have indicated the statements by the referee with "R" and our answers with "A".
R: At the point of the Anderson transition a system exhibits an "intermediate statistics", i.e. intermediate between Wigner-Dyson (ergodic phase) and Poisson (nonergodic phase). This statistics is universal, although there are several universality classes. The authors propose a correspondence between these universality classes and some "interacting topological states of matter" which occur in the Chern-Simons or string theories. As far as I can see, their logic goes like this: (i) Anderson transition is described by certain matrix ensembles. (ii) These ensembles resemble matrices which appear in some topological phases in string theory. Ergo, (iii) These very complicated phases are in correspondence with the relatively simple "intermediate statistics" of the Anderson transition (the latter is a single particle problem and in this sense it is "simple").
I have a problem with statement (i) in this logical chain. It is true that a number of authors have proposed various matrix ensembles which, supposedly, describe the Anderson transition. Those proposals, however, have little to do with the genuine Anderson transition and its intermediate statistics and, at best, can be considered as some "toy models". Indeed, the spatial dimensionality, d, plays a decisive role in the Anderson transition (the transition exists only for d>2 and the statistics is sensitive to d). But there is no "d" either in the banded matrices or in the log-Gaussian ensemble mentioned by the authors. Moreover, those ensembles don't even exhibit an Anderson transition in its usual sense, i.e. as a transition in the energy band when the energy crosses some critical value. What happens in those ensembles is that, when some parameter is changed, the eigenfunctions of the entire spectrum are changing from being localized to being extended. A simple way to see that The Anderson transition has little to do with those ensembles is simply to write down the random matrix for the original Anderson model, call it the Anderson ensemble (AE). The AE is NOT invariant under an orthogonal transformations, neither it has any similarity with the banded matrix ensemble. The main property of the AE is sparsity (the matrices in 3d are more sparse than those in 2d). Thus, I would argue that sparsity is the main property which determines the existence of the transition and the corresponding intermediate statistics.
A: We agree with the referee that Random Matrix Theory (RMT) provides only a phenomenological description of the Anderson transition. Indeed, the dimensionality is not present in the RMT description, and the class of invariant ensembles we focused on here are not related to the Anderson ensemble (AE) (which is intrinsically not invariant). Our motivation here is not to describe the Anderson model itself. Rather, we investigate a matrix model which provides a phenomenological description of certain aspects of the Anderson transition, in particular, the intermediate eigenvalue statistics, which it shares with other ergodic-to-nonergodic transitions. Indeed, as a phenomenological tool, the matrix model we consider can be applied to a very broad class of systems, for which reason it is very well known in nuclear, statistical, and condensed matter physics. On the other hand, the same matrix model was discovered in the context of high energy physics, particularly in string theory. In this context, different tools were developed and different quantities were considered than those which were thus far considered from the condensed matter perspective. The purpose of our paper is to bring those disciplines together and to allow for cross-fertilisations. Computations in our paper demonstrate this explicitly, as we apply tools and results that were developed in the context of string theory to questions in condensed matter theory.
R: In short, the whole discussion of the Anderson transition and the intermediate statistics in the paper is superficial and sometimes simply wrong. For instance, the statement "An important example of such a system is given by disordered conductors, where increasing the disorder strength leads to greater deviation from Wigner-Dyson universality" is incorrect. There is no gradual change of the Wigner-Dyson statistics with the increase of disorder. It holds all the way down to the mobility edge E_c, while below E_c the Poisson statistics sets in. The intermediate statistics takes place only at E_c (of course, one should carefully define the limiting procedure, starting with the large finite sample, where E_c is smeared into a narrow interval which however contains a macroscopic number of levels).
A: We agree that the formulation is not correct. Indeed the transition in the Anderson model is sharp. We will change the formulation in the next edition of the text.
R: All of the above doesn't imply that the paper is useless. Let's forget the Anderson transition. Then, the main content of the paper is the above statement (ii), namely, that there exists a correspondence between matrix ensembles coming from different fields, and that perhaps topological phases in string theory can be understood with the help of some simple, previously studied, matrix ensembles. This might well be true, although it's not for me to judge since I don't know anything about those topological phases in string theory.
Requested changes
At the very least, in their discussion of the Anderson localization and the matrix ensembles the authors should clearly state that those ensembles do not really describe the genuine Anderson problem but can merely serve as toy models for that phenomenon.
A: We agree with this statement and will change the formulation accordingly. Indeed, our paper is not about the Anderson transition per se - it was mentioned as one of the motivations as the matrix model we consider provides a phenomenological description of ergodic-to-nonergodic transitions which includes, but is not limited to, the Anderson transition
Anonymous on 2021-03-25 [id 1331]
In this work the authors aim to show that ergodic-to-nonergodic transitions can be analyzed using topological tools, or rather characterized in terms of quantum topological invariants. For this, they recourse on a random,matrix model formulation. More specifically, they look into an ensemble that rather uniquely appears exactly in Chern-Simons theory (not a phenomenological model in this setting, but the CS theory in question has a localization property which leads to an exact description of its observables in terms of a matrix model, reducing the complexity of a path integral) and is also a member of a family of models that posses intermediate statistics.
I think the objections regarding this second role of the model have been clarified elsewhere in the review process and I am satisfied with that.
Thus, the authors aim to exploit this somewhat unexplored venue on the dual role of the matrix model in question but for that they need to carry out an evaluation of the sff of the model, which entails computing an average in the CS matrix model not previously considered. That is what I consider to be their main achievement in the paper.
I looked with some detail into the analytical computations required and a combination of non-trivial steps involving properties of Schur polynomials are used, making the result non-trivial and, I believe, correct. At least I could not find any fault.
The discussion is thorough and accurate. The main evaluation is certainly non-trivial as it is not a mere continuation of the works on Chern-Simons matrix model or a simple use of the Schur function expansions and evaluations. It uses both of these, as it is clearly stated and referenced properly, but it requires a novel combination of the known tools, which include some crucial specific extra identity, not used before in this specific context, in Macdonald's book. For these reasons, I believe the computation of the sff to be valuable in itself and non-trivial analytically.
A great deal of material in this work is of review nature. However, this seems adequate, even necessary, due to the interdisciplinary nature of the work and the quite technical/specialized computations to obtain the main result.
How much physics one can extract from this approach perhaps remains to be fully seen, but I would say the result and the ensuing discussion is a worthy first step in this direction. The questions left open at the end are intriguing and show that further work is possible and also worth pursuing.
The only thing I would comment on, as I have not seen in their discussion, is that while they compare the sff with the CUE result, as the CS unitary matrix model manifestly goes to CUE when q-->0, I wonder why there is no discussion, seemingly, of the other limit, q--->1, where the matrix model, at least without insertions, goes in principle to GUE. In other words, their scenario seems to be one of a one-parameter extension that allows you to interpolate
CUE and GUE. That the CS matrix model is GUE when q-->1 is much more easily seen in the Hermitian matrix model formulation (e.g. the Stieltjes-Wigert model or in the original "sinh"-variables) and is also a known result in CS theory (see reviews by M. Mariño for example).
It is maybe less obvious at first sight in the unitary matrix model but it should follow, as it is elementary done, for the Gross-Witten model, for example at the beginning of Section 2 in https://arxiv.org/pdf/0704.3636.pdf . The relationship between the two formulations of CS matrix models are mentioned but not emphasized or stressed in this work (which is fine), as the authors mostly use the unitary one, but can be seen by comparing at the moments of the log-normal and the theta function (and comparing the Hankel and Toeplitz). There is also some discussion of this aspect in their reference 55 and also in the Appendix of arXiv:1103.2421.
In other words, does not the sff obtained interpolate between CUE and GUE, for q-->0 and q-->1 ? The partition functions do, but I have not explicitly considered, for the sff, the role of the trace insertions in the average. My question is simply then whether the sff is related to the sff of the GUE in the q->1 limit ? And, if not, is it because of the trace insertions and would that happen equally or differently in the two formulations of the model (Hermitian and Unitary) ?
It is not necessary to discuss this in full detail, but at least maybe some commentary on any relationship or lack thereof, eventually, with the sff of the GUE, could be interesting. Thanks.
I recommend the publication of the paper.
Author: Ward Vleeshouwers on 2021-03-30 [id 1337]
(in reply to Anonymous Comment on 2021-03-25 [id 1331])We thus far neglected to discuss how the GUE arises as the limit q -> 1, since this limit does not commute with the limit N -> infty, as was found by Onofri in doi.org/10.1007/BF02731690 . The calculation in the present work is done in the limit N -> infty; we hope to come back to the calculation of the SFF for finite N and treat the limit q -> 1 in full generality in a future work. We agree with the referee that it is worth to comment on the relationship with GUE at q -> 1, this will be added to the next edition of the present work.

---

## Round 2 · Author Response

We resubmit our work after making the revisions which were suggested by the referees.

---

## Round 2 · List of Changes

We made the revisions suggested by the referees. In particular, we corrected our previous formulation to say that the genuine Anderson localization transition is sharp, and stressed the phenomenological nature of the matrix model considered in this paper and its inability to capture the dimension-dependence that is present in the genuine Anderson model.
Further, we commented briefly on the GUE limit as q goes to 1 in the Hermitian version of the matrix model, as suggested by the second referee report. As noted in the revised version of our paper, we aim to consider this in more detail in a future work.
Lastly, we clarified a few sentences and corrected some spelling errors.

---

## Editorial Decision

published